# An Improved Multiple-Target Tracking Scheme Based on IGGM–PMBM for Mobile Aquaculture Sensor Networks

Chunfeng Lv [1], Jianping Zhu [1,*], Naixue Xiong [2,*] and Zhengsu Tao [3]

1   SOU College of Engineering Science and Technology, Shanghai Ocean University, No. 999, Huchenghuan Road, Shanghai 201306, China
2   Department of Computer, Mathematical and Physical Sciences, Sul Ross State University, Alpine, TX 79830, USA
3   Department of Electronic, Information and Electrical Engineering, Shanghai Jiaotong University, No. 800, Dongchuan Road, Shanghai 200240, China
*   Correspondence: jp-zhu@shou.edu.cn (J.Z.); neal.xiong@sulross.edu (N.X.)

**Abstract:** The Poisson multi-Bernoulli Mixture (PMBM) filter, as well as its variants, is a popular and practical multitarget tracking algorithm. There are some pending problems for the standard PMBM filter, such as unknown detection probability, random target newborn distribution, and high energy consumption for limited computational and processing capacity in sensor networks. For the sake of accommodating these existing problems, an improved multitarget tracking method based on a PMBM filter with adaptive detection probability and adaptive newborn distribution is proposed, accompanied by an associated distributed fusion strategy to reduce the computational complexities. Firstly, gamma (GAM) distribution is introduced to present the augmented state of unknown and changing target detection probability. Secondly, the intensity of newborn targets is adaptively derived from the inverse gamma (IG) distribution based on this augmented state. Then, the measurement likelihood is presented as a gamma distribution for the augmented state. On these bases, the detailed recursion and closed-form solutions to the proposed filter are derived by means of approximating the intensity of target birth and potential targets to an inverse gamma Gaussian mixture (IGGM) form and the density of existing Bernoulli components to a single IGGM form. Moreover, the associated distributed fusion strategy generalized covariance intersection (GCI), whose target states are measured by multiple sensors according to their respective fusion weights, is applied to a large-scale aquaculture tracking network. Comprehensive experiments are presented to verify the effectiveness of this IGGM–PMBM method, and comparisons with other multitarget tracking filters also demonstrate that tracking behaviors are largely improved; in particular, tracking energy consumption is reduced sharply, and tracking accuracy is relatively enhanced.

**Keywords:** multitarget point tracking; PMBM filter; GCI fusion; IGGM implementation; aquaculture





## 1. Introduction

Multiple-target tracking (MTT) can generally be defined as a process in which states and a number of spatiotemporally varying targets can be jointly determined by adopting a sequence of discrete measurements under the condition of uncertain data association, uncertain detection, uncertain process and measurement noise, random clutters, and even random newborn distribution [1,2]. Point target MTT is generally defined as tracking targets which produce at most one measurement originating from one target at each time step [3,4], while extended target MTT is generally defined as a target which can potentially occupy multiple resolution cells of one sensor, where a single extended target may produce multiple measurements in each scan step [5,6].

In this work, an improved PMBM filter is proposed to track multiple point targets. The PMBM filter provides an acceptable suboptimal approximation solution for joint estimation for the states of multiple targets, in addition to facing enormous technical

challenges [7]. Implementation of PMBM recursion mainly includes an analytic solution of a Gaussian mixture (GM) [8] and a particle solution of sequential Monte Carlo (SMC) [9,10]. SMC-based implementation does not require any assumptions about target distributions, denoting the PMBM by a sequence of random weighted particles. Nevertheless, GM-based implementation can provide a closed-form solution, which is a preferable distributed implementation for a sensor network equipped with limited processing capabilities and communication capabilities. PMBM can adopt Bayesian recursion to largely eliminate the clutter or noise originated from measurements. Furthermore, the PMBM density is a conjugate prior to both the prediction and the update processing that can preserve the PMBM form of the density, which is handy for target state density and cardinality during iterations. As a result, the PMBM filter has been increasingly adopted in many target tracking applications.

Three main contributions are presented in this paper:

1. GAM distribution is introduced to present the augmented state of unknown and changing target detection probability. The intensity of newborn targets is adaptively derived presented by IG distribution on the basis of this augmented state.
2. The measurement likelihood is presented as a gamma distribution for the augmented state. Closed-form solutions are derived on these bases by means of approximating the intensity of target birth and potential targets to an IGGM form and the density of existing Bernoulli components to a single IGGM form. Furthermore, the target cardinality distribution is estimated in the proposed filter, which is a rare solution in most PMBM filters.
3. A distributed fusion strategy GCI is applied to a large-scale aquaculture tracking network.

The remainder of this paper is structured as follows: Section 2 gives a summary of related works and the analysis premise of our model. The original PMBM filter, gamma distribution, and IG distribution, accompanied by the GCI fusion algorithm are briefly described after presenting the target dynamic model in Section 3. A novel algorithm IGGM–PMBM filter with a fusion strategy is proposed in Section 4, along with performance analysis models of IGGM–PMBM with an initialization strategy. In Section 5, accurate analyses and validations of tracking error, cardinality, and consumption are presented, and performance comparisons of the GCI–IGGM–PMBM filter with other multitarget tracking schemes are also proposed. Lastly, conclusions and discussions are presented in Section 6.

## 2. Related Works

The Poisson multi-Bernoulli mixture (PMBM) conjugate priors for MTT were studied comprehensively by Xia [11], considering the problems of both point target tracking and extended target tracking, and considering the problem of sets of objects to sets of trajectories. Some summarized studies from this thesis can be found in [6–8,10,12–14]. A derivation of the PMBM filter for multitarget tracking with the standard point target measurements without using probability generating functionals or functional derivatives was proposed in [12]. The conjugate prior of PMBM consisted of the union of a Poisson process and a multi-Bernoulli mixture (MBM), in which the MBM considered all the data association hypotheses; it could be implemented efficiently using a track-oriented multiple hypothesis tracking (MHT) formulation, while the Poisson part considered all targets that were never detected and enabled an efficient management of the number of hypotheses covering potential targets. A PMBM conjugate prior for multiple extended object filtering was proposed in [13], where a Poisson point process was used to describe the existence of yet undetected targets, while a multi-Bernoulli mixture described the distribution of the targets that were detected. The PMBM density was a conjugate prior for both the prediction and the update, which could preserve the PMBM form of the density, and a GGIW implementation was presented to approximate the unknown data associations. A new local hypothesis representation was presented in [14] to create a new Bernoulli component for each measurement, which considered computationally lighter alternatives to the extended object PMBM filter through two approximation methods. One was based on the track-oriented MB approximation, and

the other was based on the variational MB approximation via KL divergence minimization. A novel consensus-based labeled MB (LMB) filter was proposed in [15] to tackle MTT in a communication resource-sensitive distributed sensor network (DSN). Two event-triggered strategies were proposed and incorporated into the consensus-based LMB: the integral-triggering strategy (ITS) and KL divergence. A PMBM filter was proposed in [5] for coexisting point and extended targets, which was updated for a generalized measurement model including measurements originated from point and extended targets. A single-target space accommodated both point and extended targets, and then derived the filtering recursion propagating Gaussian densities for point targets and GGIW densities for extended targets. Considering the above, the PMBM filter can be adopted in multitarget tracking filters, which can provide relatively optimal solutions for point targets or extended targets.

A robust PMBM filter using adaptive birth distributions for the tracking of multiple targets was proposed in [16], which presented a novel measurement-driven adaptive birth distribution robust to the random locations where new targets are located. Beta distribution was employed to describe unknown detection probability to improve the method practicability. The detailed recursion and closed-form solutions were derived through two approximations: one approximating the target birth intensity and potential target intensity to the beta GGIW (BGGIW) mixture form, and the other approximating the existing Bernoulli component density to a single BGGIW form. The Gibbs sampler was adopted to resolve the problem of the computational bottleneck caused by data associations. A novel fusion framework for the Poisson MB (PMB) filter was proposed in [17], which integrated both the advantages of the TOMB/P filter in dealing with missed detection and the advantages of the MOMB/P filter in dealing with coalescence. The Bernoulli components in different MB distributions were associated with each other by KLD minimization to fuse the different PMB distributions.

A standard PMBM or PHD (CPHD) filter assumes that the target birth intensity is known a priori, but newborn targets (including spontaneous newborns and spawned newborns) may be randomly located in the coverage range with adaptive newborn density in actual applications, which leads to primitive missed detection and inaccurate estimation. An improved GM–CPHD filter was developed in [18] to estimate target cardinality distribution of the time-varying newborns at each processing step adopting a discrete kernel estimator in conjunction with an exponential weighted moving average scheme. Target birth intensity could be updated according to the outputting estimated birth cardinality distribution, and predicted birth intensity and cardinality distribution could be adopted by a tracker based on GM–CPHD to adjust its filtering strength for target tracking. Novel extensions were derived to distinguish between the persistent and the newborn targets in both the prediction and the update step in [19], allowing the PHD/CPHD filter to adapt the target birth intensity at each processing step using the received measurements. This measurement-driven birth intensity is practical because it removes the need for the prior specification of birth intensities and eliminates the restriction of target appearance volumes within state space. As described in [16], BGGIW was adopted to approximate target birth intensity and potential target intensity.

The target detection probability depends on the sensor, target, environment, and features used for detection. A priori knowledge of target detection probability is of critical importance in a PMBM or PHD (CPHD) filter, while the detection probability is always time-varying and space-varying. Target amplitude and SNR information were exploited to present the detection probability in [20], and the IGGM distribution was used to implement the PHD and CPHD filters to address the limitation of nonnegativity of the target amplitude and SNR and non-Gaussian noise. A variational Bayesian approximation method was brought into the trajectory PHD (VB–TPHD) filter to obtain measurement noise covariance adaptively in [21]. The random matrix, following the IW distribution, was introduced to model the unknown measurement covariance in this filter. The VB–TPHD filter minimized the KLD and estimated the series of states for multiple targets, taking noise covariance matrices into account simultaneously. An R-PMBM recursion that estimates the unknown

detection probability was proposed in [4], in which the state of each object was coupled with a variable representing the detection probability such that the standard PMBM filtering process evolved into an R-PMBM filter which could jointly estimate the state of objects and detection probability. A beta function was proposed to represent the detection probability for a computationally feasible implementation of the R-PMBM filter. The beta function was also adopted to describe the unknown detection probability in [16].

Other tracking methods exist, such as energy-based autoregressive neural network [22,23], deep learning method [24–26], visual method [27,28], and genetic algorithm [29]. A novel transfer learning algorithm with an SMC–PHD filter and GM–PHD filter was proposed in [30] to automatically adapt the YOLO network with unlabeled target sequences. The frames of the unlabeled target sequences were automatically labeled. The detection probability and clutter density of the SMC–PHD filter and GM–PHD were applied to retrain the YOLO network for occluded targets and clutter. A novel likelihood density with the confidence probability of the YOLO detector and visual context indications was implemented to choose target samples. A simple resampling strategy was proposed for SMC–PHD YOLO to address the weight degeneracy problem.

A novel consensus CPHD filter based on GM implementation was presented for distributed MTT over a sensor network [31]. A fusion robust fusion method was proposed to perform fusion via fully distributed means. Tracking information (including location and other tracking information) of multiple targets measured by separate nodes was fused for the practically more relevant case of unknown correlations between nodes. The fusion method was implemented on only two agents or sensors; one fusion weight was $w$ and the other was $1 - w$. A sequential GCI fusion means was proposed in [32] after applying a GGIW–CPHD filter to each sensor, which also assumed that there were two agents or sensors to be fused. A pairwise means was adopted if there were more than two sensors; hence, if there were more than two sensors, every two sensors were coupled to one pairwise function to fuse the tracking information using the fusion means of [19]. This pairwise fusion could substantially decrease the computational burden compared to the batch fusion, as well as bring about some problems. One is that it needed to establish location correlations before coupling these sensors, which consumed much energy for the uncertainty location of mobile sensors. The other is that one residual sensor could not be coupled with any other sensor after coupling with its corresponding sensor.

As described above, a Bayesian recursion adopted PMBM filter can largely improve the tracking behaviors by eliminating the clutter or noise originating from measurements, but there remain several challenging tasks. Firstly, a standard PMBM filter assumes that the birth density is a priori known density or homogenous density [19], while new targets can birth or die at arbitrary positions or at random process period in real-world application scenarios, leading to PMBM becoming inapplicable. Secondly, the PMBM filter assumes that target detection probability is a priori known, while detection probability can arise with uncertain spatial and uncertain temporal distribution, leading to PMBM becoming incomplete and inaccurate. Thirdly, many measurements detected by a mass of sensors can bring about a sharp increase in computation and processing burden, which is a pending problem for sensor networks. Lastly, tracking initialization is not included in most PMBM filters, leading to tracking error and increased energy consumption in tracking, or leading to increased cumulative tracking errors in subsequent steps. In order to solve these existing problems of the standard PMBM filter, an improved multitarget tracking method based on a GCI–IGGM–PMBM filter is proposed in this work for our aquaculture tracking sensor network. Firstly, a GAM distribution is introduced to present the augmented state of unknown and changing target detection probability. Secondly, the intensity of newborn targets is adaptively derived presented using the IG distribution based on this augmented state. Then, the measurement likelihood is presented as a gamma distribution for the augmented state. On these bases, the detailed recursion and closed-form solutions to the proposed filter are derived by means of approximating the intensity of target birth and potential targets to an IGGM form and the density of existing Bernoulli components to

a single IGGM form. Moreover, the associated distributed fusion strategy GCI, whose target states are measured by multiple sensors according to their respective fusion weights, is applied to a large-scale aquaculture tracking network. Lastly, the target states are initialized by adopting the hidden terminal couple (HTC) scheme [33] in the proposed system. Furthermore, comprehensive experiments are presented to verify the effectiveness of the GCI–IGGM–PMBM tracking method. Moreover, comparisons with other multitarget tracking schemes also demonstrate that the tracking behavior is largely improved; in particular, the tracking energy consumption is reduced sharply, and the tracking accuracy is relatively enhanced.

## 3. System Models

Figure 1 shows the physical nodes in our aquaculture network, detecting the multi-target location and other tracking information. Sensors (including normal sensing nodes and anchors in charge of information processing and fusion) are randomly located in a $(-500, 500) \times (-500, 500)$ m$^2$ area, according to a two-dimensional Poisson distribution with a density of $\lambda_0$, in which the ratio of anchors is $\gamma_0$, i.e., $N_A(1 - hop) = \pi \gamma_0 \lambda_0$. Multiple targets, Chinese crabs in this study, move along with maneuvering trajectories in the sensor deployment area and are detected by the sensor nodes (normal sensors or anchors) implementing the IGGM–PMBM tracking strategy.

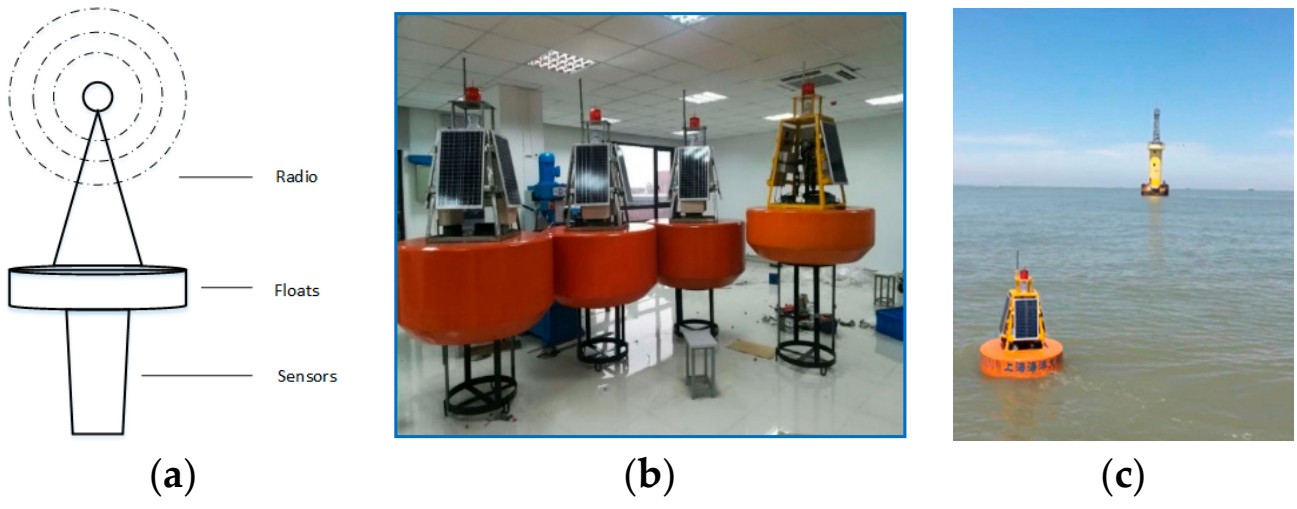

**(a)** **(b)** **(c)**

**Figure 1.** The nodes in the aquaculture network of tracking targets adopted in the environment for our aquaculture sensor network. (**a**) Schematic diagram of the structural representation of nodes. (**b**) Buoys equipped with sensors in this network. (**c**) Layout representation of tracking system in this aquaculture network.

Multiple targets, equipped with recognizable sensors, can also communicate with other sensors in this sensor network. Each node can establish its one-hop and two-hop neighbor lists through an HTC scheme. In each timestep, nodes including targets can perform the HTC scheme to obtain its one-hop and two-hop neighbors, before initializing the filter. Each HTC request packet contains the node ID, location, and current moment. Receiving this request packet, nodes (including targets) can obtain distance information of their one-hop and two-hop neighbors. Consecutive position information detected from the request packet of specific targets is used to obtain the targets' velocity. Location and velocity information can be used for the iterative IGGM–PMBM filter to obtain target tracking.

### 3.1. Multitarget Bayes Filter

For limited-energy large-scale sensor networks, a method with less computational cost and less complexity for target tracking is anticipated. The target state is $X_k$ (matrices are denoted in italic uppercase nonbold letters in this study) at timestep $k$ within the state space

**X** (sets are denoted in uppercase bold letters), which contains kinematic states (position, velocity, turn rate, orientation, etc.). Measurements at timestep $k$ are represented as $M_k$ within the measurement space **M**, which denotes all measurement sets from time $t = 1$ to $k$, including time $t = k$. The state and measurement set of the targets are represented by $X_k \in F(X)$ and $M_k \in F(M)$, which are the finite subsets of $X$ and $M$, respectively. The system model and measurement model can be generally described as follows [4]:

$$X_k = \left\{ x_k^1, x_k^2, \ldots, x_k^{N_k} \right\} \in F(X), \tag{1}$$

$$M_k = \left\{ m_k^1, m_k^2, \ldots, m_k^{Z_k} \right\} \in F(M), \tag{2}$$

where $N_k$ and $Z_k$ are the number of targets and the number of measurements at timestep $k$, respectively. Target state $x_k$ at time $k$ contains three parts, which are targets surviving from the previous timestep $k - 1$, newborn targets spawned from targets at the previous timestep $k - 1$, and spontaneous newborn targets at the timestep $k$, which are indicated as $S_{k|k-1}$, $SP_{k|k-1}$, and $B_k$, respectively; $X_k = S_{k|k-1} \cup SP_{k|k-1} \cup B_k$. It is worth noting that newborn targets spawned from targets at the previous timestep $k - 1$ can be considered as newborn targets for classification and modeling simplicity, which is denoted as $X_k = S_{k|k-1} \cup B_k$.

The multitarget state posterior density at time can be denoted as $p_{k|k}(X_k|M_{1:k})$. According to Bayesian estimation theory, multitarget state posterior density can be obtained by optimal Bayes recursion as follows:

$$p_{k|k-1}(X_k|M_{1:k-1}) = \int f_{k|k-1}(X_k|X_{k-1}, M_{1:k-1}) p_{k-1|k-1}(X_{k-1}|M_{1:k-1}) dX_{k-1}, \tag{3}$$

$$p_{k|k}(X_k|M_{1:k}) = \frac{l_k(M_k|X_k) p_{k|k-1}(X_k|M_{1:k-1})}{\int l_k(M_k|X_k) p_{k|k-1}(X_k|M_{1:k-1}) dX_k}, \tag{4}$$

where $f_{k|k-1}(X_k|X_{k-1}, M_{1:k})$ is the state transition density function based on all measurement sets $M_{1:k}$ from $M_1$ at time $t = 1$ up to $M_k$ at time $t = k$ and target state $X_{k-1}$ at previous time $t = k - 1$, and $l_k(M_k|X_k)$ denotes the target measurement likelihood function.

*3.2. PMBM RFS*

The PMBM filter is a combination of two disjointed parts, the Poisson point process (PPP) and the multi-Bernoulli mixture (MBM). The PPP describes the distribution of the targets that exist but are not detected, which is expressed by $X^u$. The MBM describes the mixture distribution of the targets that have been detected at least once, represented by a series of weighted multi-Bernoulli density, expressed as $X^d$.

$$X^u \cup X^d = X, X^u \cap X^d = \varnothing, \tag{5}$$

$$p_k(X^u) = e^{-\langle D_k(\mathbf{x}); 1 \rangle} \prod_{X \in \mathbf{X}^u} D_k(X) = e^{-\lambda} \prod_{X \in \mathbf{X}^u} \lambda f_k(X), \tag{6}$$

$$p_k(X^d) = \sum_{\cup_{i \in \mathbb{I}} \mathbf{X}^i = \mathbf{X}^d} \prod_{i \in \mathbb{I}} p_k^i(X^i), \tag{7}$$

$$p_k^i(X^i) = \begin{cases} 1 - r_k^i & X^i = \varnothing \\ r_k^i p_k^i(x) & X^i = \{x\}, \\ 0 & |X^i| \geq 2 \end{cases} \tag{8}$$

$$p_k(X_k) = \sum_{\mathbf{X}^u \cup \mathbf{X}^d = \mathbf{X}} p_k(X_k^u) \sum_{j \in \mathbb{J}} w_k^j p_k^j(X_k^d) = \sum_{\mathbf{X}^u \cup \mathbf{X}^d = \mathbf{X}} e^{-\int D_k(X^u) dx} \prod_{x \in \mathbf{X}^u} D_k(X^u) \sum_{j \in \mathbb{J}} \sum_{\cup_{i \in \mathbb{I}} \mathbf{X}^i = \mathbf{X}^d} \prod_{i \in \mathbb{J}} w_k^{j,i} r_k^{j,i} p_k^{j,i}(X_k^{j,i}), \tag{9}$$

where $p_k^{j,i}(X_k^{j,i})$ represents the density of the $i$th Bernoulli in the $j$th MB, $\mathbb{I}$ is an index set of independent Bernoulli components, and $\mathbb{J}$ is an index set for the MBs in the MBM or com-

ponents of the MBM. $w_k^{j,i}$ is the probability of the *i*th Bernoulli component density in the *j*th global hypothesis. Then, the density of PMBM can be presented as shown in Equation (9), which is represented entirely by the parameters of $D_k(X^u)$, $\left\{ w_k^j, \left\{ r_k^{j,i}, p_k^{j,i}(X^{j,i}) \right\}_{i \in \mathbb{I}^j} \right\}_{j \in \mathbb{J}}$. Because the density of PMBM is a prior MTT conjugate, the PMBM density is a prior conjugate with both the prediction and the update, preserving the PMBM form of the density.

### 3.3. PMBM Recursion

The recursion of PMBM consists of the prediction and updating step.

Prediction Step: Given a posterior PMBM density of $p_k(X_k)$ at timestep k, as shown in Equation (9) with the parameters of $D_k(X_k^u)$, $\left\{ w_k^j, \left\{ r_k^{j,i}, p_k^{j,i}(X_k^{j,i}) \right\}_{i \in \mathbb{I}_k^j} \right\}_{j \in \mathbb{J}_k}$ and the standard dynamic model, the predicted density is the PMBM density expressed in Equation (10).

$$p_{k+1}(X_{k+1}) = \sum_{X^u \cup X^d = \mathbf{X}} e^{-\int D_{k+1}(X^u)dX} \prod_{x \in \mathbf{X}^u} D_{k+1}(X^u) \sum_{j \in \mathbb{J}_{k+1}} w_{k+1}^j \sum_{\cup_{i \in \mathbb{I}_{k+1}} \mathbf{X}^i = \mathbf{X}^d} \prod_{i \in \mathbb{I}_{k+1}^j} r_{k+1}^{j,i} p_{k+1}^{j,i}(X^{j,i}). \tag{10}$$

$$D_{k+1}(X^u) = D_{k+1}^b(X) + \int D_k(X^u) p_s f_{k+1|k}(X) dX. \tag{11}$$

$$r_{k+1}^{j,i} = r_k^{j,i} \int p_s p_k^{j,i}(X) dX. \tag{12}$$

$$p_{k+1}^{j,i}(X^{j,i}) = \frac{\int p_k^{j,i}(X) p_s f_{k+1|k}(X) dX}{\int p_s p_k^{j,i}(X) dX}. \tag{13}$$

$$w_{k+1}^j = \sum_{i \in \mathbb{I}^j} w_k^{j,i}. \tag{14}$$

Updating Step: Given a prior PMBM density of $p_{k+1}(X_{k+1})$ at timestep $k+1$, as shown in Equation (10) with the parameters of $D_{k+1}(X_{k+1}^u)$, $\left\{ w_{k+1}^j, \left\{ r_{k+1}^{j,i}, p_{k+1}^{j,i}(X_{k+1}^{j,i}) \right\}_{i \in \mathbb{I}_{k+1}^j} \right\}_{j \in \mathbb{J}_{k+1}}$ and a set of measurements $\mathbf{M}$, the updated density is the PMBM density expressed in Equation (15).

$$p_{k+1}(X_{k+1}|M_k) = \sum_{X^u \cup X^d = \mathbf{X}} p_k^u(X_k^u) \sum_{j \in \mathbb{J}_{k+1}} \sum_{A \in \mathcal{A}^j} w_{A,k}^j p_{A,k}^j(X_k^d), \tag{15}$$

$$p_k^u(X_k^u) = e^{-\int D_k(X^u)dX} \prod_{x \in \mathbf{X}^u} D_k(X^u), \tag{16}$$

$$p_{A,k}^j(X_k^d) = \sum_{\cup_{C \in A} \mathbf{X}^C = \mathbf{X}^d} \prod_{C \in A} r_{C,k}^j p_{C,k}^j(X^C), \tag{17}$$

$$w_{A,k}^j = \frac{w_{k+1}^j \prod_{C \in A} \mathcal{L}_C}{\sum_{j \in \mathbb{J}_{k+1}} \sum_{A \in \mathcal{A}^j} w_{k+1}^j \prod_{C \in A} \mathcal{L}_C}, \tag{18}$$

$$D_k(X^u) = q_D(X^u) D_{k+1}(X^u), \tag{19}$$

$$\mathcal{L}_C = \begin{cases} \kappa_{C_C} + \int l_{C_C} D_{k+1}^u(X)dX & \text{if } C \cap \mathbb{I}^j = \varnothing, \ |C_C| = 1 \\ \int l_{C_C} D_{k+1}^u(X)dX & \text{if } C \cap \mathbb{I}^j = \varnothing, \ |C_C| > 1 \\ 1 - r_{k+1}^{j,i_C} + r_{k+1}^{j,i_C} \int q_D p_{k+1}^{j,i_C}(X)dX & \text{if } C \cap \mathbb{I}^j \neq \varnothing, \ |C_C| = \varnothing \\ r_{k+1}^{j,i_C} \int l_{C_C} p_{k+1}^{j,i_C}(X)dX & \text{if } C \cap \mathbb{I}^j \neq \varnothing, \ |C_C| \neq \varnothing \end{cases} \tag{20}$$

$$r_{C,k}^{j} = \begin{cases} \dfrac{\int D_{k+1}^{u}(X)l_{C_C}dX}{\kappa_{C_C} + \int D_{k+1}^{u}(X)l_{C_C}dX} & if\ C \cap \mathbb{I}^{j} = \varnothing,\ |C_C| = 1 \\ 1 & if\ C \cap \mathbb{I}^{j} = \varnothing,\ |C_C| > 1 \\ \dfrac{\int D_{k+1}^{u}(X)l_{C_C}dX}{1 - r_{k+1}^{j,i_C} + r_{k+1}^{j,i_C}\int q_D p_{k+1}^{j,i_C}(X)dX} & if\ C \cap \mathbb{I}^{j} \neq \varnothing,\ C_C = \varnothing \\ 1 & if\ C \cap \mathbb{I}^{j} = \varnothing,\ C_C \neq \varnothing \end{cases}, \tag{21}$$

$$p_{C,k}^{j}(X^C) = \begin{cases} \dfrac{D_{k+1}^{u}(X)l_{C_C}}{\int D_{k+1}^{u}(X)l_{C_C}dX} & if\ C \cap \mathbb{I}^{j} = \varnothing \\ \dfrac{q_D(X)p_{k+1}^{j,i_C}(X)}{\int q_D(X)p_{k+1}^{j,i_C}(X)dX} & if\ C \cap \mathbb{I}^{j} \neq \varnothing, C_C = \varnothing \\ \dfrac{l_{C_C}(X)p_{k+1}^{j,i_C}(X)}{\int l_{C_C}p_{k+1}^{j,i_C}(X)dX} & if\ C \cap \mathbb{I}^{j} \neq \varnothing, C_C \neq \varnothing \end{cases}, \tag{22}$$

where $q_D$ is the probability of missed detection, and $\mathcal{L}_C$ is the likelihood function of data association $C$. $\mathcal{A}^j$ represents all data associations $A$ for the predicted global hypothesis of the $j$th MB, and $C \in A$ represents an assignment of each measurement cell $MC$ in **M** to an existing target, either to a new target or to a clutter.

### 3.4. The Inverse Gamma Distribution and Gamma Distribution

Gamma: The probability density of the Gamma distribution $\mathcal{G}(x; \alpha, \beta)$ is presented as

$$\mathcal{GAM}(\zeta; \alpha, \beta) = \frac{\beta^\alpha}{\Gamma(\alpha)}\zeta^{\alpha-1}e^{-\beta\zeta}, \tag{23}$$

where shape parameter $\alpha > 0$ and rate parameter $\beta > 0$, and $\Gamma(\alpha)$ denotes the gamma function. Its mode and mean are $(\alpha-1)/\beta$ and $\alpha/\beta$, respectively.

Inverse Gamma: The probability density of the inverse gamma distribution $\mathcal{IG}(x; \alpha, \beta)$ is defined over the support $x > 0$ as

$$\mathcal{IG}(\eta; \alpha, \beta) = \frac{\beta^\alpha}{-(\alpha)}\eta^{-\alpha-1}\exp(-\frac{\beta}{\eta}), \tag{24}$$

where shape parameter $\alpha > 0$ and rate parameter $\beta > 0$. The mode in which the probability density function is the maximum is $\beta/(\alpha+1)$, and the mean value is $\beta/(\alpha-1)$. The variance of the IG distribution is $\beta^2/[(\alpha-1)^2(\alpha-2)]$.

### 3.5. GCI Fusion

In our aquaculture sensor network, multiple sensors (including normal nodes and anchors) are sensing, exchanging, and processing location, state, and other tracking information from targets and their neighbors. Suitable information fusion strategies are taken into account in such an energy-limited system with limited sensing capabilities, limited processing capabilities, and limited computation capabilities. The centralized fusion methods cannot be adopted for such large-scale sensors and large monitoring areas. The pairwise fusion means cannot be adopted; although it can substantially decrease the computational burden, it brings about some problems. One is that it needs to establish location correlations before coupling these sensors, which consumes great energy for the uncertainty location of mobile sensors. The other is that one residual sensor cannot couple with any other sensor after coupling with its corresponding sensor, leading to missed detection. Thus, a distributed fusion method based on GCI, a multitarget fusion strategy, is proposed in this sensor network to iterate local fusion among neighboring nodes for scalability requirement. The GCI fusion process is briefly described below [31]. Given estimates $\hat{x}_k$ of the state $x$

from multiple estimators with relative covariances $P_k$ and unknown correlations, the GCI fusion can be presented as

$$GCI(x) = \frac{\prod_k [p_k(x)]^{w_k}}{\int \prod_k [p_k(x)]^{w_k} dx}, \tag{25}$$

where $p_k(x) \sim N(x; \hat{x}_k, P_k)$ is the Gaussian PDF with mean $\hat{x}_k$ and covariance $P_k$, and $w_k$ is the weight of $x_k$. Of course, the fusion of PDF in Equation (25) can be used for an arbitrary PDF, not only for a Gaussian PDF.

Given that the target RFS set follows an independent and identically distributed cluster process [30], the sensor multitarget densities to be fused to take the following form:

$$GCI_k(X) = |X|! p_k(X) \prod_{x \in X} D_k(x), \tag{26}$$

$$\overline{GCI}_k(X) = |X|! \overline{p}_k(X) \prod_{x \in X} \overline{D}_k(x). \tag{27}$$

As related above, $D_k(x)$ and $N_k(x, n)$ are considered the state density and cardinality density of multiple targets, respectively. The GCI fusion of $D_k(x)$ and $N_k(x, n)$ of the multiple targets adopting Equations (25) and (26) can be obtained as follows:

$$\overline{D}(x) = \frac{\prod_{i=1}^{N_A} [D_k(x)]^{w_i}}{\int \prod_{i=1}^{N_A} [D_k(x)]^{w_i} dx}, \tag{28}$$

$$\overline{N}(x, n) = \frac{\prod_{i=1}^{N_A} [N_i(n)]^{w_i} \left\{ \int \prod_{i=1}^{N_A} [D_k(x)]^{w_i} dx \right\}^n}{\sum_{j=1}^{\infty} \prod_{i=1}^{N_A} [N_k(j)]^{w_i} \left\{ \int \prod_{i=1}^{N_A} [D_k(x)]^{w_i} dx \right\}^j}. \tag{29}$$

GCI fusion of $\overline{D}(x)$ expressed in Equation (28) describes that the target state density $\overline{D}(x)$ at time $k$ is the weighted geometric mean of target state densities $D(x)$, and the fusion cardinality $\overline{N}(x, n)$ expressed in Equation (29) is an interconnected mean not only with the target state density but also with the target cardinality density. For practical implementation, GCI fusions of $\overline{D}(x)$ and $\overline{N}(x, n)$ can be considered as finite dimensional parameters, such as the target number of $n_{\max} = N_k$ in $\overline{N}(x, n)$. Two finite parameterized methods are most commonly adopted to process the infinite dimensional problem of target state density, SMC and GM. For our computation-limited and process-limited sensor network, the GM approach is adopted, which promises to be more applicable. Generally, the order of magnitude of the involved number of Gaussian components is lower than the number of particles of the SMC, which is required for applicable tracking performance.

During GM implementation for the fusion of Equation (28), the GM state densities can be obtained as

$$D(x) = \sum_{u=1}^{N_A} w^u N(x; \hat{x}^u, P^u), \tag{30}$$

$$D_k(x) = \sum_{u=1}^{N_{A,k}} (w_k{}^u) N(x; \hat{x}_k^u, P_k^u), \tag{31}$$

where $w^u$ is the measured state weight by the $u$th anchor, and $N_A$ is the number of anchors which are responsible for sensing, processing, and fusing state information. Then, the target state fusion can be obtained using Equation (31). It is worth noting that our aquaculture sensor network is composed of more than two sensors. The sequential pairwise fusion method described in [31] can be modified to multiple sensors via associated distributed fusion. Associated fusion refers to the associative fusion of target states measured by multiple sensors according to their respective weights $w^u$ and $\sum_{u=1}^{N_A} w^u = 1$. Sensors can

obtain their own neighbor lists at the beginning of each timestep *k*, and the relationship of measured target states can be established.

$$\overline{D}_f(\mathbf{x}) = \frac{\prod\limits_{u=1}^{N_A} [D_u(\mathbf{x})]^{w^u}}{\int \prod\limits_{u=1}^{N_A} [D_u(\mathbf{x})]^{w^u} dx}. \tag{32}$$

According to the simplified integral algorithm of the power of Gaussian components [31], Equation (32) can be expressed as Equation (33), in which $\eta(w^u, P^u)$ can be expressed as Equation (34).

$$\left[\sum_{u=1}^{N_A} \alpha^u N(x; \hat{x}^u, P^u)\right]^{w^u} \cong \sum_{u=1}^{N_A} [\alpha^u N(x; \hat{x}^u, P^u)]^{w^u} = \sum_{u=1}^{N_A} (\alpha^u)^{w^u} \eta(w^u, P^u) N(x; \hat{x}^u, \frac{P^u}{w^u}). \tag{33}$$

$$\eta(w^u, P^u) = \frac{[\det(2\pi P^u w^{-1})]^{1/2}}{[\det(2\pi P^u)]^{w_u/2}}. \tag{34}$$

## 4. Our Proposed GCI–IGGM–PMBM Scheme

Multitarget tracking for aquaculture sensor networks allows obtaining the state density and cardinality density of multiple culture objects adopting an appropriate highly efficient tracking method. According to the standard multitarget PMBM model in Section 3.2, some pending problems should be considered.

First of all, detection probability is of critical importance in a multitarget tracking scheme, and the standard PMBM filter assumes that the detection probability is known and time-invariant. In actual multitarget tracking, targets can move in uncertain environments at different times, and sensors collecting target state information can also be located in an uncertain area of tracking position. Thus, the detection probability has much uncertainty and may be time-variant, which brings about a decrease in the accuracy of target state or cardinality depending on the detection measurements.

Secondly, the standard PMBM filter tracks multiple targets with the assumption that newborn target birth density is known and time-invariant, or the prior density is known [19], which only takes the known observation area into target density estimation without considering the unknown observation area. However, newborn targets (including spontaneous newborns and spawned newborns) may be randomly located in the coverage range with adaptive newborn density in actual applications, leading to primitive PMBM with further missed detection and inaccurate estimation.

Lastly, large-scale sensors located in the tracking coverage range can collect and exchange location information originating from multiple targets or other tracking information for target tracking. For such an energy-limited, computation-limited, and processing-limited sensor network, it is impossible for each sensor, even if it is an anchor with better hardware equipment, to execute and process so much tracking information. In addition to increased energy consumption for tracking multiple targets, extensive delays are also taken into account for exchanging and processing tracking data by the respective sensors.

In order to solve these existing problems of the standard PMBM filter, an improved method based on a GCI–IGGM–PMBM filter is proposed for our aquaculture tracking sensor network. Firstly, an adaptive target detection probability, modeled as a gamma distribution, is taken into account to estimate multiple target states and the cardinality density. Secondly, an adaptive newborn density originating from measurements based on the time-variant detection probability is proposed, with IG distribution. Lastly, a GCI fusion strategy, consisting of an associative distributed fusion of target states measured by multiple sensors according to their respective weights, is applied to a large-scale tracking network.

This improved GCI–IGGM–PMBM scheme estimates the target state density and cardinality of multiple targets, as elaborately presented below.

### 4.1. Augmented State Model

A variable $\xi \in [0,1]$, which represents the unknown detection probability in a tracking system, is augmented into the multitarget state $X$. That is, the new multitarget states can be presented as $\hat{X} = (X, \xi)$, and the integral of the density of the augmented state $\hat{X}$ can be denoted as

$$\int p(\hat{X})d\hat{X} = \int \int_0^1 p(X, \xi)d\xi dX. \tag{35}$$

The state transition density $f_{k|k-1}(\hat{X}_k|\hat{X}_{k-1})$ of the augmented state $\hat{X}_k$ given the state $\hat{X}_{k-1}$ can be denoted as

$$f_{k|k-1}(\hat{X}_k|\hat{X}_{k-1}) = f_{k|k-1}(X_k, \xi_k|X_{k-1}, \xi_{k-1}) = f_{k|k-1}(X_k|X_{k-1})\hat{f}_{k|k-1}(\xi_k|\xi_{k-1}), \tag{36}$$

$$l_k(M_k|X_{\hat{}}) = l_k(M_k|X_k, \xi_k) = l_k(M_k|X_k), \tag{37}$$

where $f_{k|k-1}(X_k|X_{k-1})$ is the transition density of the state $X_k$ given the state $X_{k-1}$, and $\hat{f}_{k|k-1}(\xi_k|\xi_{k-1})$ is the transition density of variable $\xi_k$ given variable $\xi_{k-1}$. Then, the survival probability $p_{S,k}(\hat{X}_k)$ and detection probability $p_{D,k}(\hat{X}_k)$ can be presented as

$$p_{S,k}(\hat{X}_k) = p_{S,k}(X_k, \xi_k) = p_{S,k}(X_k), \tag{38}$$

$$p_{D,k}(\hat{X}_k) = p_{D,k}(X_k, \xi_k) = \xi_k. \tag{39}$$

### 4.2. Recursion Based on Augmented States

In the proposed GCI–IGGM–PMBM filter, the newborn intensity and detection probability are a priori unknown, which is unlike the standard PMBM filter. Gamma distribution is used to present the unknown detection probability, and target states comply with Gaussian distribution. The prediction and updating of IGGM–PMBM can be derived as described below.

Proposition 1 (Predicted process). Given the posterior intensity of Poisson $D_{k-1}(\hat{\mathbf{X}}^u, \xi)$, the newborn intensity with $B_k^b(\hat{\mathbf{X}}_b, \xi)$ and MBM RFS with parameters $\left\{ w_{k-1}^j, \left\{ r_{k-1}^{j,i}, p_{k-1}^{j,i}(\hat{\mathbf{x}}) \right\}_{i\in\mathbb{I}^j} \right\}_{j\in\mathbb{J}}$ at time $k-1$, representing the undetected targets and potentially detected targets, respectively, the predicted intensity of IGGM–PMBM can be obtained in two steps, as described below.

1. PPP process

$$D_{k|k-1}^u(X_k, \xi_k) = B_k^b(X_b, \xi_k) + D_{k-1}^u(X_k, \xi_k), \tag{40}$$

$$B_k^b(X_b, \xi_k) = \int \int_0^1 B_{k-1}^b(X_{k-1}^b, \xi_{k-1}^b) p_S(X_{k-1}^b) f_{k|k-1}(X_k|X_{k-1}^b) \hat{f}_{k|k-1}(\xi_k|\xi_{k-1}) dX_{k-1}^b d\xi_{k-1}^b, \tag{41}$$

$$D_{k-1}^u(X_k, \xi_k) = \int \int_0^1 D_{k-1}^u(X_{k-1}, \xi_{k-1}) p_S(\xi_{k-1}) f_{k|k-1}(X_k|X_{k-1}) \hat{f}_{k|k-1}(\xi_k|\xi_{k-1}) dX_{k-1} d\xi_{k-1}, \tag{42}$$

where $f_{k|k-1}(X_k|X_{k-1}^b)$ denotes the transition function of target states, and $\hat{f}_{k|k-1}(\xi_k|\xi_{k-1})$ denotes the transition function of the augmented variate, presented as an unknown detection probability.

2. MBM process

$$r_{k|k-1}^{j,i} = r_{k-1}^{j,i} \int \int_0^1 p_{k-1}^{j,i}(X_{k-1}, \xi_{k-1}) p_s(\xi_{k-1}) dX_{k-1} d\xi_{k-1}, \tag{43}$$

$$p_{k|k-1}^{j,i}(X,\xi) = \int \int_0^1 p_{k-1}^{j,i}(X_{k-1},\xi_{k-1})p_s(\xi_{k-1})f_{k|k-1}(X_k|X_{k-1})\hat{f}_{k|k-1}(\xi_k|\xi_{k-1})dX_{k-1}d\xi_{k-1}, \tag{44}$$

$$w_{k|k-1}^j(X,\xi) = w_{k-1}^j(X). \tag{45}$$

Then, the predicted density is also a PMBM density, which is represented by the parameters $D_{k|k-1}^u(X_k,\xi_k)$ and $\left\{w_{k|k-1}^j, \left\{r_{k|k-1}^{j,i}, p_{k|k-1}^{j,i}(\hat{x})\right\}_{i\in\mathbb{I}^j}\right\}_{j\in\mathbb{J}}$.

Proposition 2 (Updating process). Given the predicted PMBM density with parameters $D_{k|k-1}^u(X_k,\xi_k)$ and $\left\{w_{k|k-1}^j, \left\{r_{k|k-1}^{j,i}, p_{k|k-1}^{j,i}(\hat{x})\right\}_{i\in\mathbb{I}^j}\right\}_{j\in\mathbb{J}}$, as well as the measurements $M_k$ at time $k$, the updated density of IGGM–PMBM can be obtained in four stages as described below.

1. Update for undetected targets

$$p_{k|k}(X,\xi) = (1-\xi)p_{k|k-1}(X,\xi). \tag{46}$$

2. Update for potential targets detected for the first time

$$r_{k,1}^u = \frac{\int_0^1 \int D_{k|k-1}^u(X_{k-1},\xi_{k-1})l_k(M_k|X_{k-1})dX_{k-1}d\xi_{k-1}}{\kappa_C(M_k,\xi_{k-1}) + \int_0^1 \int D_{k|k-1}^u(X_{k-1},\xi_{k-1})l_k(M_k|X_{k-1})dX_{k-1}d\xi_{k-1}}, \tag{47}$$

$$p_{k,1}^u(X,\xi|M) = \frac{l_k(M|X)D_{k|k-1}^u(X,\xi)}{\int_0^1 \int l_k(M_k|X_{k-1})D_{k|k-1}^u(X_{k-1},\xi_{k-1})dX_{k-1}d\xi_{k-1}}, \tag{48}$$

$$w_{k,1}^j = \frac{w_{k|k-1}^j \prod_{C\in A_1}\left(\kappa_C(M_k,\xi_{k-1}) + \int_0^1 \int D_{k|k-1}^u(X_{k-1},\xi_{k-1})l_k(M_k|X_{k-1})dX_{k-1}d\xi_{k-1}\right)}{\sum_{j\in\mathbb{J}}\sum_{A_1\in\mathcal{A}^j} w_{k|k-1}^j \prod_{C\in A_1}\left(\kappa_C(M_k,\xi_{k-1}) + \int_0^1 \int D_{k|k-1}^u(X_{k-1},\xi_{k-1})l_k(M_k|X_{k-1})dX_{k-1}d\xi_{k-1}\right)}, \tag{49}$$

where $\kappa_C(M_k,\xi_{k-1})$ denotes the clutter density originating from measurements based on the augmented variate unknown detection probability, and the subscript 1 denotes the update type.

3. Missed detection of MBM

$$r_{k,2}^{j,i} = \frac{r_{k|k-1}^{j,i}\int_0^1 \int D_{k|k-1}^{j,i}(X_{k-1},\xi_{k-1})(1-\xi_{k-1})dX_{k-1}d\xi_{k-1}}{1 - r_{k|k-1}^{j,i} + r_{k|k-1}^{j,i}\int_0^1 \int D_{k|k-1}^u(X_{k-1},\xi_{k-1})(1-\xi_{k-1})dX_{k-1}d\xi_{k-1}}, \tag{50}$$

$$p_{k,2}^{j,i}(X,\xi|M) = \frac{(1-\xi_k)D_{k|k-1}^u(X,\xi)}{\int_0^1 \int (1-\xi_k)D_{k|k-1}^u(X_{k-1},\xi_{k-1})dX_{k-1}d\xi_{k-1}}, \tag{51}$$

$$w_{k,2}^j = \frac{w_{k|k-1}^j \prod_{C\in A_2}\left(1 - r_{k|k-1}^{j,i} + r_{k|k-1}^{j,i}\int_0^1 \int D_{k|k-1}^u(X_{k-1},\xi_{k-1})(1-\xi_{k-1})dX_{k-1}d\xi_{k-1}\right)}{\sum_{j\in\mathbb{J}}\sum_{A_2\in\mathcal{A}^j} w_{k|k-1}^j \prod_{C\in A_2}\left(1 - r_{k|k-1}^{j,i} + r_{k|k-1}^{j,i}\int_0^1 \int D_{k|k-1}^u(X_{k-1},\xi_{k-1})(1-\xi_{k-1})dX_{k-1}d\xi_{k-1}\right)}, \tag{52}$$

where the definitions of $\mathbb{J}$ and $\mathcal{A}^j$ are the same as in Section 3.1.

4. Update for MBM

$$r_{k,3}^{j,i} = 1, \tag{53}$$

$$p_{k,3}^{j,i}(X,\xi|M) = \frac{l_k(M|X)p_{k|k-1}^{j,i}(X,\xi)}{\int_0^1 \int l_k(M_k|X_{k-1})p_{k|k-1}^{j,i}(X_{k-1},\xi_{k-1})dX_{k-1}d\xi_{k-1}}, \tag{54}$$

$$w_{k,3}^j = \frac{w_{k|k-1}^j \prod_{C\in A_3} r_{k|k-1}^{j,i}\int_0^1 \int p_{k|k-1}^{j,i}(X_{k-1},\xi_{k-1})l_k(M_k|X_{k-1})dX_{k-1}d\xi_{k-1}}{\sum_{j\in\mathbb{J}}\sum_{A_3\in\mathcal{A}^j} w_{k|k-1}^j \prod_{C\in A_3} r_{k|k-1}^{j,i}\int_0^1 \int p_{k|k-1}^{j,i}(X_{k-1},\xi_{k-1})l_k(M_k|X_{k-1})dX_{k-1}d\xi_{k-1}}, \tag{55}$$

where $l_k(M_k|X_{k-1})$ denotes the measurement likelihood.

*4.3. IGGM Implementation*

The closed-form solution using adaptive newborn distributions accompanied by augmented unknown detection probability can be derived using an inverse gamma Gaussian mixture. The gamma, inverse gamma, and Gaussian distributions are presented as the unknown detection probability, adaptive newborn distribution, and target state distribution, respectively. The representations for gamma and inverse gamma distributions were described in Section 3.4, and the recursion of IGGM–PMBM is proposed below. It is assumed that the IGG component at time $k-1$ is given as

$$p_{k|k}(X,\xi) = \mathcal{GAM}(\xi_\xi;\alpha_{\xi,k-1},\beta_{\xi,k-1}) \times \mathcal{IG}(X,\xi_B;\alpha_{B,k-1},\beta_{B,k-1})\mathcal{N}(X,\xi_B;m_{B,k-1},P_{B,k-1}) \times \mathcal{N}(X;m_{x,k-1},P_{x,k-1}), \quad (56)$$

$$l_k(M|X,\xi) = \mathcal{GAM}(\xi_\xi;\alpha_{\xi,k-1},\beta_{\xi,k-1})\mathcal{N}(M,\xi;H_kX,R_k). \quad (57)$$

The adaptive newborn distribution and target state Gaussian distribution can be simplified as IGG; then, the IGG prediction can be derived as

$$p_{k|k-1}(X,\xi) = \mathcal{GAM}(\xi_{k|k-1};\alpha_{\xi,k|k-1},\beta_{\xi,k|k-1}) \times \mathcal{IGG}(X,\xi;\alpha_{B,k|k-1},\beta_{B,k|k-1},m_{B,k|k-1},P_{B,k|k-1},m_{x,k|k-1},P_{x,k|k-1}), \quad (58)$$

where $\alpha_{\xi,k|k-1} = \frac{\alpha_{k-1}}{T_\xi}$, $\beta_{\xi,k|k-1} = \frac{\beta_{k-1}}{T_\xi}$, $\alpha_{B,k|k-1} = \frac{\alpha_{k-1}}{T_B}$, $\beta_{B,k|k-1} = \frac{\beta_{k-1}}{T_B}$, $m_{B,k|k-1} = Fm_{B,k-1}$, $P_{B,k|k-1} = FP_{B,k-1}F^T + Q_{B,k|k-1}$, $m_{x,k|k-1} = Fm_{x,k-1}$, and $P_{x,k|k-1} = FP_{x,k-1}F^T + Q_{x,k|k-1}$. $T_\xi$ and $T_B$ denote the measurement rate parameters for adaptive detection probability and adaptive newborn probability, respectively. $F, Q_{B,k|k-1}, Q_{x,k|k-1}$ denotes the transition matrix, process covariance for newborn distribution, and process covariance for the target state. Then, the IGG updating can be derived as

$$p_k(X,\xi) = \mathcal{GAM}(\xi_k;\alpha_{\xi,k},\beta_{\xi,k}) \times \mathcal{IGG}(X;\alpha_{B,k},\beta_{B,k},m_{B,k},P_{B,k},m_{x,k},P_{x,k}), \quad (59)$$

where $\alpha_{\xi,k} = \alpha_{\xi,k|k-1} + |M_k|$, $\beta_{\xi,k} = \beta_{\xi,k|k-1} + 1$, $\alpha_{B,k} = \alpha_{B,k|k-1} + |B_k|$, $\beta_{B,k} = \beta_{B,k|k-1} + 1$, $m_{B,k} = m_{B,k|k-1} + K\varepsilon$, $P_{B,k} = P_{B,k|k-1} - KHP_{B,k|k-1}$, $m_{x,k} = m_{x,k|k-1} + K\varepsilon$, and $P_{x,k} = P_{x,k|k-1} - KHP_{x,k|k-1}$.

$H$ denotes the measurement matrix, and $K,\varepsilon$ denote the new observations or measurements based on state transition. The intensity of the newborn target can be assumed to be an IGGM form as follows:

$$\begin{aligned} D_{k-1}^B(X,\xi) &= \sum_{l=1}^{J_{k-1}^B} w_{k-1}^{l,B}\mathcal{IG}(X,\xi_B;\alpha_{B,k-1}^{l,B},\beta_{B,k-1}^{l,B})\mathcal{G}(X;m_{B,k-1}^{l,B},P_{B,k-1}^{l,B}) \\ &= \sum_{l=1}^{J_{k-1}^B} w_{k-1}^{l,B}\mathcal{IGG}(X,\xi_B;\alpha_{B,k-1}^{l,B},\beta_{B,k-1}^{l,B},m_{B,k-1}^{l,B},P_{B,k-1}^{l,B}) \end{aligned} \quad (60)$$

Therefore, the intensity of the Poisson process at time $k-1$ can also be presented as an IGGM form:

$$D_{k-1}^u(X,\xi) = \sum_{l=1}^{J_{k-1}^u} w_{k-1}^{l,u}\mathcal{GAM}(\xi_\xi;\alpha_{\xi,k-1}^{l,u},\beta_{\xi,k-1}^{l,u})\mathcal{IGG}(X,\xi;\alpha_{B,k|k-1}^{l,u},\beta_{B,k|k-1}^{l,u},m_{B,k|k-1}^{l,u},P_{B,k|k-1}^{l,u},m_{X,k|k-1}^{l,u},P_{X,k|k-1}^{l,u}). \quad (61)$$

The density of MBM in an IGGM form with parameters $\left\{w_{k-1}^j, \left\{r_{k-1}^{j,i}, p_{k-1}^{j,i}(X_{k-1}^{j,i},\xi_{k-1}^{j,i})\right\}_{i\in\mathbb{I}}\right\}_{j\in\mathbb{J}}$ can be presented as

$$p_{k-1}^{j,i}(X,\xi) = \mathcal{GAM}(\xi_\xi;\alpha_{\xi,k-1}^{j,i},\beta_{\xi,k-1}^{j,i})\mathcal{IGG}(X,\xi;\alpha_{B,k-1}^{j,i},\beta_{B,k-1}^{j,i},m_{B,k-1}^{j,i},P_{B,k-1}^{j,i},m_{X,k-1}^{j,i},P_{X,k-1}^{j,i}). \quad (62)$$

**Proposition 3** (Predicted process). Given that the intensity of Poisson process is in an IGGM form and the density of the *ith* Bernoulli component in the *jth* hypothesis is in a single IGG

form, the predicted intensity of the Poisson and Bernoulli components of the MBM process are presented as described below.

1.  PPP process

$$
\begin{aligned}
D_{k|k-1}^{u}(X,\xi) = &\sum_{l_1=1}^{J_{k-1}^{B}} w_{k|k-1}^{l_1,B} \mathcal{IGG}(X,\xi_B; \alpha_{k|k-1}^{l_1,B}, \beta_{k|k-1}^{l_1,B}, m_{k|k-1}^{l_1,B}, P_{k|k-1}^{l_1,B}) \\
&+ \sum_{l_2=1}^{J_{k-1}^{u}} w_{k|k-1}^{l_2,u} \mathcal{GAM}(\xi_\xi; \alpha_{k|k-1}^{l_2,u}, \beta_{k|k-1}^{l_2,u}) \mathcal{IGG}(X,\xi_u; \alpha_{k|k-1}^{l_2,u}, \beta_{k|k-1}^{l_2,u}, m_{k|k-1}^{l_2,u}, P_{k|k-1}^{l_2,u})
\end{aligned} \tag{63}
$$

$$
w_{k|k-1}^{l_1,B} = p_{S,k|k-1} w_{k-1}^{l_1,B}, w_{k|k-1}^{l_1,u} = p_{S,k|k-1} w_{k-1}^{l_1,u}, \tag{64}
$$

2.  MBM process

$$
w_{k|k-1}^{j,i} = w_{k-1}^{j,i}, \tag{65}
$$

$$
r_{k|k-1}^{j,i} = p_{S,k} r_{k-1}^{j,i}, \tag{66}
$$

$$
p_{k|k-1}^{j,i}(X,\xi) = \mathcal{GAM}(\xi_\xi; \alpha_{\xi,k|k-1}^{j,i}, \beta_{\xi,k|k-1}^{j,i}) \mathcal{IGG}(X,\xi; \alpha_{B,k|k-1}^{j,i}, \beta_{B,k|k-1}^{j,i}, m_{B,k|k-1}^{j,i}, P_{B,k|k-1}^{j,i}, m_{X,k|k-1}^{j,i}, P_{X,k|k-1}^{j,i}), \tag{67}
$$

$$
m_{B,k|k-1}^{j,i} = F_{k-1} m_{B,k-1}^{j,i}, P_{B,k|k-1}^{j,i} = Q_{k-1} + F_{k-1} P_{B,k-1}^{j,i} F_{k-1}^{T}. \tag{68}
$$

Proposition 4 (Updating process). Given that the predicted intensity of the Poisson process is in an IGGM form, as shown in Equation (67), where $J_{k|k-1}^{u} = J_{k-1}^{B} + J_{k-1}^{u}$, the predicted density of the MBM can be expressed as shown in Equation (69).

$$
D_{k|k-1}^{u}(X,\xi) = \sum_{l=1}^{J_{k|k-1}^{u}} w_{k|k-1}^{l,u} \mathcal{GAM}(\xi_\xi; \alpha_{k|k-1}^{l,u}, \beta_{k|k-1}^{l,u}) \mathcal{IGG}(X,\xi_u; \alpha_{k|k-1}^{l,u}, \beta_{k|k-1}^{l,u}, m_{k|k-1}^{l,u}, P_{k|k-1}^{l,u}), \tag{69}
$$

$$
p_{k|k-1}^{j,i}(X,\xi) = \mathcal{GAM}(\xi_\xi; \alpha_{\xi,k|k-1}^{j,i}, \beta_{\xi,k|k-1}^{j,i}) \mathcal{IGG}(X,\xi; \alpha_{B,k|k-1}^{j,i}, \beta_{B,k|k-1}^{j,i}, m_{B,k|k-1}^{j,i}, P_{B,k|k-1}^{j,i}, m_{X,k|k-1}^{j,i}, P_{X,k|k-1}^{j,i}). \tag{70}
$$

Furthermore, the measurement $M_k$, the update of the Poisson process, and the MBM process can be derived in four stages.

1.  Update for undetected targets

$$
D_{k}^{u}(X,\xi) = \sum_{l=1}^{J_{k|k-1}^{u}} w_{k1}^{l,u} \mathcal{GAM}(\xi_\xi; \alpha_{k1}^{l,u}, \beta_{k1}^{l,u}) \mathcal{IGG}(X,\xi_u; \alpha_{k1}^{l,u}, \beta_{k1}^{l,u}, m_{k1}^{l,u}, P_{k1}^{l,u}), \tag{71}
$$

where $\alpha_{k1}^{l,u} = \alpha_{k|k-1}^{l,u}$, $\beta_{k1}^{l,u} = \beta_{k|k-1}^{l,u}$, $m_{k1}^{l,u} = m_{k|k-1}^{l,u}$, $P_{k1}^{l,u} = P_{k|k-1}^{l,u}$, and $w_{k1}^{l,u} = w_{k|k-1}^{l,u} \left( \frac{\beta_{k|k-1}^{l,u}}{\beta_{k|k-1}^{l,u}+1} \right)^{\alpha_{k|k-1}^{l,u}}$.

2.  Update for potential targets detected for the first time

$$
r_{k,2}^{2}(M) = \frac{\displaystyle\sum_{l=1}^{J_{k|k-1}^{u}} w_{k|k-1}^{l,u} \mathcal{G}(M,\xi_k; H_k m_{k|k-1}^{l,u}, H_k P_{k|k-1}^{l,u} H_k^T + R_k)}{\kappa_C(M_k,\xi_k) + \displaystyle\sum_{l=1}^{J_{k|k-1}^{u}} w_{k|k-1}^{l,u} \mathcal{G}(M,\xi_u; H_k m_{k|k-1}^{l,u}, H_k P_{k|k-1}^{l,u} H_k^T + R_k)}, \tag{72}
$$

$$
p_{k,2}(X,\xi|M) = \frac{\displaystyle\sum_{l=1}^{J_{k|k-1}^{u}} w_{k|k-1}^{l,u} \mathcal{GAM}(M,\xi_k; \alpha_{k2}^{l,u}, \beta_{k2}^{l,u}) \mathcal{IGG}(M,\xi_u; \alpha_{k2}^{l,u}, \beta_{k2}^{l,u}, m_{k2}^{l,u}, P_{k2}^{l,u}, H_k m_{k|k-1}^{l,u}, H_k P_{k|k-1}^{l,u} H_k^T + R_k)}{\displaystyle\sum_{l=1}^{J_{k|k-1}^{u}} w_{k|k-1}^{l,u} \mathcal{G}(M,\xi_u; H_k m_{k|k-1}^{l,u}, H_k P_{k|k-1}^{l,u} H_k^T + R_k)}, \tag{73}
$$

where $\alpha_{k2}^{l,u} = \alpha_k^{l,u}$, $\alpha_{k2}^{l,u} = \alpha_k^{l,u}$, $m_{k2}^{l,u} = m_{k|k-1}^{l,u} + P_{k|k-1}^{l,u} H_k (H_k^T P_{k|k-1}^{l,u} H_k^T + R_k)^{-1}$ $(M - H_k m_{k|k-1}^{l,u})$, and $P_{k2}^{l,u} = (I - P_{k|k-1}^{l,u} H_k^T (H_k P_{k|k-1}^{l,u} H_k^T + R_k)^{-1} H_k) P_{k|k-1}^{l,u}$.

3. Missed detection of MBM

$$r_{k,3}^{j,i} = r_{k|k-1}^{j,i}, w_{k,3}^{j,i} = w_{k|k-1}^{j,i} \left( \frac{\beta_{k|k-1}^{j,i}}{\beta_{k|k-1}^{j,i} + 1} \right)^{\alpha_{k|k-1}^{j,i}}, \tag{74}$$

$$p_{k,3}^{j,i}(X, \xi) = \mathcal{GAM}(\xi; \alpha_{k3}^{j,i}, \beta_{k3}^{j,i}) \mathcal{IGG}(\xi; \alpha_{k3}^{j,i}, \beta_{k3}^{j,i}, m_{k3}^{j,i}, P_{k3}^{j,i}), \tag{75}$$

where $\alpha_{k3}^{j,i} = \alpha_{k|k-1}^{j,i}$, $\beta_{k3}^{j,i} = \beta_{k|k-1}^{j,i} + 1$, $m_{k3}^{j,i} = m_{k|k-1}^{j,i}$, and $P_{k3}^{j,i} = P_{k|k-1}^{j,i}$.

4. Update for MBM

$$r_{k,4}^{j,i}(M) = 1, \tag{76}$$

$$p_{k,4}^{j,i}(X, \xi|M) = \mathcal{GAM}(\xi; \alpha_{k4}^{j,i}, \beta_{k4}^{j,i}) \mathcal{IGG}(\xi; \alpha_{k4}^{j,i}, \beta_{k4}^{j,i}, m_{k4}^{j,i}, P_{k4}^{j,i}), \tag{77}$$

$$w_{k,4}^{j,i}(M) = w_{k|k-1}^{j,i} r_{k|k-1}^{j,i} \mathcal{G}(M, \xi; H_k m_{k|k-1}^{j,i}, H_k P_{k|k-1}^{j,i} H_k^T + R_k), \tag{78}$$

where $\alpha_{k4}^{j,i} = \alpha_{k|k-1}^{j,i} + |M_k|$, $\beta_{k4}^{j,i} = \beta_{k|k-1}^{j,i} + 1$, $m_{k4}^{j,i} = m_{k|k-1}^{j,i} + P_{k|k-1}^{l,u} H_k^T (H_k P_{k|k-1}^{l,u} H_k^T + R_k)^{-1} (M - H_k m_{k|k-1}^{j,i})$, and $P_{k4}^{j,i} = (I - P_{k|k-1}^{l,u} H_k^T (H_k P_{k|k-1}^{l,u} H_k^T + R_k)^{-1} H_k) P_{k|k-1}^{j,i}$.

### 4.4. Fusion

Given the posterior density of multitarget states $D_k(x)$ derived from a sensor with fusion weight $w^u$, the fused result of their IGGM state density and cardinality density can be approximated as shown in Equations (79) and (80), respectively.

$$\overline{D}_{f(x,k)}(X, \xi) = \left\{ \sum_{i=1}^{J_{B,k}+J_{x,k}} w_k \mathcal{IGG}(X, \xi; \alpha_k^i, \beta_k^i, m_k^i, P_k^i) \right\}^{W_j}$$
$$\cong \sum_{i=1}^{J_{B,k}+J_{x,k}} w_k^{W_j} [(\mathcal{GAM}(X, \xi; \alpha_k^i, \beta_k^i)]^{W_j} [\mathcal{IG}(X, \xi; \alpha_k^i, \beta_k^i)]^{W_j} [\mathcal{G}(X, \xi; m_k^i, P_k^i)]^{W_j} \tag{79}$$

$$\overline{D}_{f(x,k)}(x, \xi; m) = \overline{D}_{f(B,k)}(x, R) \otimes I_{B,k} + \overline{D}_{f(PMBM,k)}(x, R) \otimes I_{PMBM,k}, \tag{80}$$

where $\otimes$ denotes the Kronecker product, and $I_{B,k}$ and $I_{PMBM,k}$ denote the identity matrix of dimensional $D_{B,k}$ and $D_{PMBM,k}$, respectively. The fusion cardinality can be expressed as shown in Equation (81).

$$\overline{N}(n) = \frac{\prod_k [N_k(n)]^{w_k} \left\{ \int \prod_k [D_k(x)]^{w_k} dx \right\}^n}{\sum_{j=1}^{N_A} \prod_i [N_k(j)]^{w_k} \left\{ \int \prod_k [D_k(x)]^{w_k} dx \right\}^j}. \tag{81}$$

The estimation sets of state density and cardinality can be computed according to true measurements and corresponding noise. The GCI-IGGM-PMBM scheme is shown in Algorithm 1.

---

**Algorithms 1** A framework of GCI-IGGM-PMBM algorithms

---

Step 1: Initialization

    For $k = 1$, $N_x = 0$,

        Adopt HTC scheme and establish one-hop and two-hop neighbor lists;

        then, obtain the location information at $k = 1$.

    At timestep $k - 1$;

    Input: $\xi_\xi, \xi_B, \alpha_{\xi,k-1}, \beta_{\xi,k-1}, \alpha_{B,k-1}, \beta_{B,k-1}, m_{B,k-1}, P_{B,k-1}, m_{x,k-1}, P_{x,k-1}; M_k$;

    Output: $p_{k|k}(X, \xi), p_k(X, \xi), p_{k-1}^{j,i}(X, \xi), \updownarrow_k(M|X, \xi), D_{k-1}^B(X, \xi), D_{k-1}^u(X, \xi)$.

Step 2: Prediction

Prediction for PPP for time step $k - 1$:

    Input: $\xi_B, \xi_u; \zeta_\xi; \alpha_{k|k-1}^{l_1,B}, \beta_{k|k-1}^{l_1,B}, m_{k|k-1}^{l_1,B}, P_{k|k-1}^{l_1,B}; \alpha_{k|k-1}^{l_2,u}, \beta_{k|k-1}^{l_2,u}, m_{k|k-1}^{l_2,u}, P_{k|k-1}^{l_2,u}; w_{k|k-1}^{l_1,B}, w_{k|k-1}^{l_2,u};$
$\left\{ w_{k-1}^j, \left\{ r_{k-1}^{j,i}, p_{k-1}^{j,i}(\hat{\mathbf{x}}) \right\}_{i \in \mathbb{I}^j} \right\}_{j \in \mathbb{J}};$

    Output: $D_{k|k-1}^u(X, \xi)$;

    for $j = 1 : N_{S,k} + N_{B,k}$,

        use Equations (63) and (64);

    end for

Prediction for survival for timestep $k - 1$:

    Input: $\xi_\xi; \alpha_{\xi,k|k-1}^{j,i}, \beta_{\xi,k|k-1}^{j,i}; \alpha_{B,k|k-1}^{j,i}, \beta_{B,k|k-1}^{j,i}, m_{B,k|k-1}^{j,i}, P_{B,k|k-1}^{j,i}, m_{X,k|k-1}^{j,i}, P_{X,k|k-1}^{j,i}$;

    Output: $p_{k|k-1}^{j,i}(X, \xi)$;

    for $j = 1 : N_{S,k} + N_{B,k}$,

        use Equations (65)–(68);

    end for

Step 3: Update based on augmented variable for timestep $k$:

Update for undetected targets,

    Input: $w_{k1}^{l,u}, \xi_\xi, \xi_u; \alpha_{k1}^{l,u}, \beta_{k1}^{l,u}, m_{k1}^{l,u}, P_{k1}^{l,u}$;

    Output: $D_k^u(X, \xi)$;

    for $j = 1 : N_{S,k} + N_{B,k}$,

        use Equation (71);

    end for

Update for potential targets detected for the first time,

    Input: $w_{k|k-1}^{l,u}; \xi_\xi, \xi_u; \alpha_{k2}^{l,u}, \beta_{k2}^{l,u}, m_{k2}^{l,u}, P_{k2}^{l,u}; M_k; m_{k|k-1}^{l,u}, P_{k|k-1}^{l,u}$;

    Output: $r_{k,2}^2(M), p_{k,2}(X, \xi|M)$;

    for $j = 1 : N_{S,k} + N_{B,k}$,

        use Equations (72) and (73);

    end for

Missed detection of MBM,

    Input: $w_{k|k-1}^{j,i}; \xi_k, \xi_u; \alpha_{k2}^{l,u}, \beta_{k2}^{l,u}, m_{k2}^{l,u}, P_{k2}^{l,u}; M_k; m_{k|k-1}^{l,u}, P_{k|k-1}^{l,u}$;

    Output: $w_{k,3}^{j,i}, r_{k,3}^2(M), p_{k,3}^{j,i}(X, \xi)$;

    for $j = 1 : N_{S,k} + N_{B,k}$,

        use Equations (74) and (75);

    end for

Update for MBM,

    Input: $w^j; \xi; \alpha_{k4}^{l,u}, \beta_{k4}^{l,u}, m_{k4}^{l,u}, P_{k4}^{l,u}$;

    Output: $w_{k,4}^{j,i}(M), r_{k,4}^j(M), p_{k,4}^{j,i}(X, \xi|M)$;

    for $j = 1 : N_{S,k} + N_{B,k}$,

        use Equations (76)–(78);

    end for

Step 4: Fusion

Input: $\xi; J_{B,k}, J_{x,k}; w_k, W_j; \alpha_k^i, \beta_k^i, m_k^i, P_k^i$;

Output: $\overline{D}_{f(x,k)}(X, \xi), \overline{N}(n)$;

    for $j = 1 : N_{S,k} + N_{B,k}$,

        use Equations (79)–(81)'

end for

Step 5: State extraction

The state estimation set at time $k$: $D_k(x, \xi) = D_{B,k}(x, \xi) + D_{PMBM,k}(x, \xi)$.

---

## 5. Performance Analysis

The proposed GIC–IGGM–PMBM filter provides an applicable method to combine adaptive newborn density with adaptive detection probability based on target states following a Gaussian distribution. In this section, comprehensive simulations are presented to validate the accuracy of this proposed method, using the NS-3 and MATLAB simulator according to previous analyses. A total of 100 Monte Carlo trials are performed for each simulation, and the results represent the average simulation results obtained. The simulation setup is similar to that of the MC-MPMC [34].

Target states and measurements can be presented as $\mathbf{x}_k = F_{k|k-1}\mathbf{x}_{k-1} + G_k\mathbf{q}_{k-1}$ and $\mathbf{m}_k = H_k\mathbf{x}_k + R_k$, where $\mathbf{x}_k$ is the state vector at time $k$ denoted as $\mathbf{x}_k = (x_k\ \dot{x}_k\ y_k\ \dot{y}_k\ \omega_k)^T$ (vectors are denoted in lowercase bold letters), $x_k$ and $y_k$ are the states of target position, $\dot{x}_k$ and $\dot{y}_k$ are the states of target velocity, and $\omega_k$ is the turn rate. $F_k$ is the target transition matrix, $G_k$ is the control input matrix, and $\mathbf{q}_{k-1}$ is the process noise which follows a Gaussian distribution with zero mean and covariance $Q_k$, defined as $E[q_k q_k^T]$. $\mathbf{m}_k$ is the measurement vector at time $k$ denoted as $\mathbf{m}_k = (x_k\ y_k\ \omega_k)^T$, $H_k$ is the measurement matrix, and $R_k$ is the measurement noise. The process noise and measurement noise are mutually uncorrelated to each other. The parameter values can be presented as the values in [34].

Assuming that there are 12 targets following their separate trajectory in the simulations, true trajectories of these targets are as shown in Figure 2. The birth time and death time of all targets with the initial states are shown in Table 1. The processing time slot was set to $T = 1s$, $\sigma_x = 0.01$, $\sigma_y = 0.01$, and $\sigma_\omega = (\pi/180)^2$. The clutter generated in each measurement was assumed to abide by a uniform distribution [16]. The optimal sub-pattern assignment (OSPA) [35] and generalized OSPA (GOSPA) were introduced to comprehensively evaluate the tracking behaviors of different schemes or those of different parameters originating from the same scheme with $p = 1$ and $c = 200$. At first, we performed the proposed GCI–IGGM–PMBM filter, and then compared its tracking performance with the BGM–PMBM [4], IGGM–PHD (CPHD) [20], and BGGIW–PMBM [16] filters based on unknown detection probability. Next, the tracking behavior comparisons between GCI–IGGM–PMBM and BGGIW–PMBM are presented as a function of the characteristic parameters of adaptive newborn distribution.

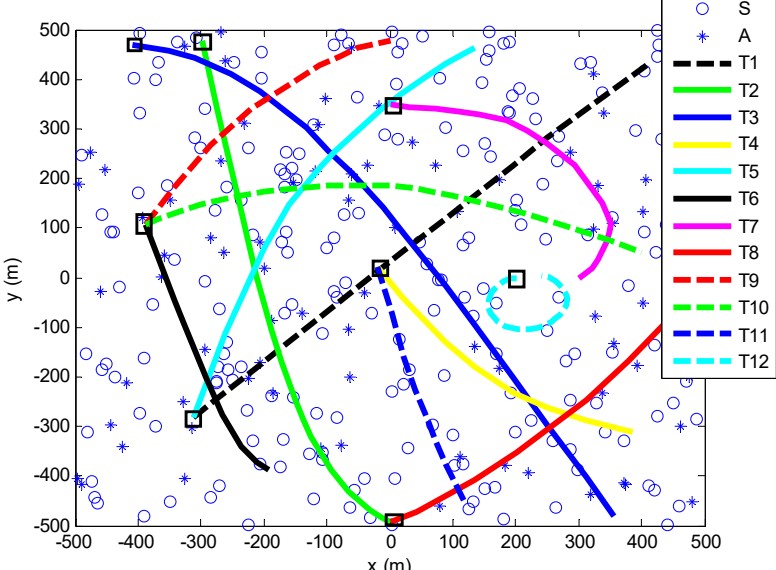

**Figure 2.** Target trajectories: small squares represent locations of target birth, and the line destination point represents the locations of target death, which are not labeled. The labels of *S*, *A*, and *Tn* are abbreviations of the normal sensor, anchor, and trajectory n, respectively.

**Table 1.** The initialization of true targets, along with their birth time and death time.

| Target Index | Initialization Targets States | Birth Time (s) | Death Time (s) |
|---|---|---|---|
| Target 1 | $[-310, -280, 5.23, 4.12, 0]$ | 1 | 100 |
| Target 2 | $[-300, 480, 4.12, -3.23, -2.78]$ | 20 | 100 |
| Target 3 | $[-410, 470, 3.45, -4.32, 0.41]$ | 58 | 100 |
| Target 4 | $[0, 0, 2.74, -4.54, -0.56]$ | 15 | 78 |
| Target 5 | $[-310, -280, 3.65, 3.56, 0.61]$ | 37 | 90 |
| Target 6 | $[-390, 110, 5.32, -2.35, -2.21]$ | 1 | 50 |
| Target 7 | $[20, 350, 4.23, -4.12, 3.02]$ | 45 | 80 |
| Target 8 | $[0, -490, 3.23, 5.23, -0.34]$ | 67 | 92 |
| Target 9 | $[-388, 108, 3.54, 3.37, 0.43]$ | 1 | 100 |
| Target 10 | $[-388, 108, 1.32, 0.32, 0.75]$ | 1 | 30 |
| Target 11 | $[-19, 22, 2.12, -4.72, -4.27]$ | 67 | 100 |
| Target 12 | $[202, 0, -0.37, -1.53, -0.23]$ | 45 | 90 |

*5.1. Validations of Unknown Detection Probability*

Figures 3–8 show the performance comparisons between the GCI–IGGM–PMBM filter and other multitarget tracking filters based on unknown detection probability. Figure 3a–d present the position estimations of the BGM–PMBM, IGGM–PHD, BGGIW–PMBM, and IGGM–PMBM filters, respectively. Overall, the trajectories of these filters mainly abided by the true trajectories, and the tracking estimations were accurate. Several different demonstrations in terms of the details are shown in Figure 3. Trajectory 2 (T2) and Trajectory 12 (T12) presented relatively large position errors for the BGM–PMBM filter, as shown in Figure 3a. T2 offset its trajectory from the initial tracking step and accumulated this error, whereas T12 presented poor turning performance for its bearing sets with a small turn rate. T1 and T3 for the IGGM–PHD filter presented relatively large errors, as shown in Figure 3b, especially for T3. T5 and T12 for the BGGIW–PMBM filter presented poor tracing performance; its computational burden was relatively high, and it took no account of bearing or turn rate in its designs. The IGGM–PMBM filter presented relatively superior tracking behaviors compared to the three other filters, as can be seen from the simulation results for several factors. For example, the IGGM–PMBM filter could work at an arbitrary turn rate. Performing the distributed fusion strategy using the global distributions of gamma, inverse gamma, and PMBM components, other than MB components, could substantially save energy.

The actual but unknown detection probabilities were set to $p_D = 0.7$, $p_D = 0.9$, and $p_D = 0.96$. The clutter rate was $\lambda_c = 10$, and the anchor rate was $\gamma_0 = 0.1$. The OSPA distances for different $p_D$ were relatively high for the periods of 26–34 s, 46–53 s, 68–74 s, and 88–92 s, as shown in Figure 4a–c, which represent the intersecting points of these trajectories. The lowest OSPA distances for GCI–IGGM–PMBM at the initial trajectories prove that its HTC exchange strategy was applicable and accurate in the tracking initialization step. The distance error was relatively high at a lower detection probability; for example, the error for $p_D = 0.7$ was much higher than that for $p_D = 0.96$, and the error was highest for $p_D = 0.7$.

The cardinality estimations for the periods of intersecting points, i.e., at 26–34 s, 46–53 s, 68–74 s, and 88–92 s, exhibited great deviation from the true trajectories shown in Figure 5. With an increase in the detection probability, the tracking performance improved. The cardinality estimations of GCI–IGGM–PMBM followed the true trajectory with relatively high accuracy.

The detection probability plays an important role in multitarget tracking performance. For the processing period, 100 s in our simulations, the detection probability was changing and adaptive. The estimations of average OSPA error, cardinality, and $p_D$ are shown in Figure 6. The value $p_{D,k}$ was $p_D = 0.95$ at $k \in (0, 17]$, $p_D = 0.9$ at $k \in (17, 37]$, $p_D = 0.85$ at $k \in (37, 48]$, $p_D = 0.8$ at $k \in (48, 71]$, $p_D = 0.85$ at $k \in (71, 82]$, and $p_D = 0.7$ at $k \in (82, 100]$. Overall, the estimations for OSPA error, cardinality, and detection probability mainly abided by the true values, and tracking estimations were accurate, as shown in Figure 6. On the other hand, abrupt changes occurred at the change junctions of the detection probability, e.g., at 17–18 s, 37–38 s, 48–49 s, 71–72 s, and 82–83 s. All filters containing the GCI–

IGGM–PMBM filter could take some time to adjust according to the changes in detection probability, which led to greater OSPA errors or relative higher inconformity of cardinality.

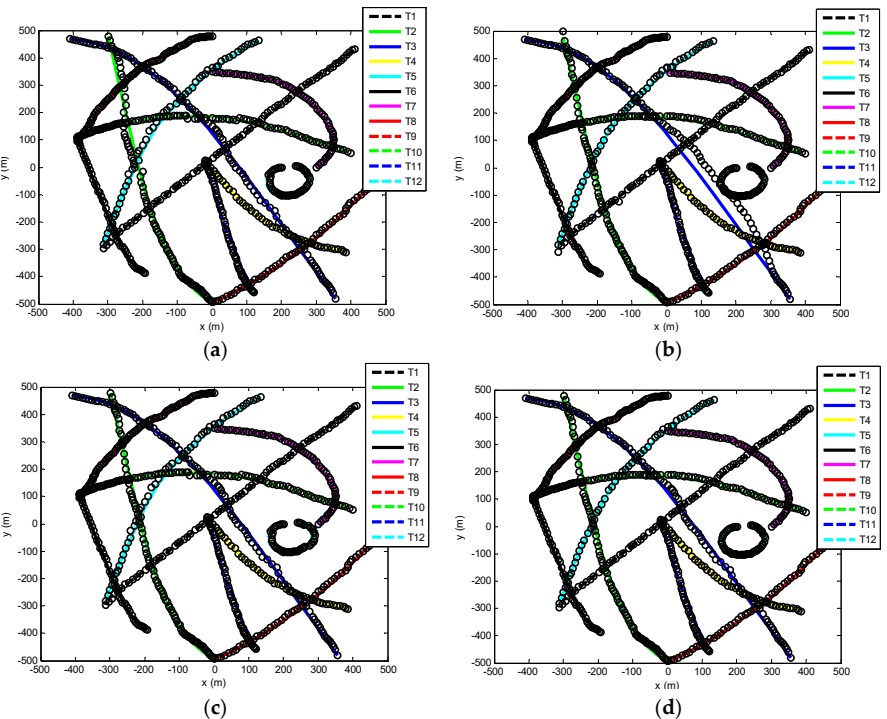

**Figure 3.** Position estimation: black circles are the real track points. (**a**) BGM–PMBM; (**b**) IGGM–PHD; (**c**) BGGIW–PMBM; (**d**) IGGM–PMBM.

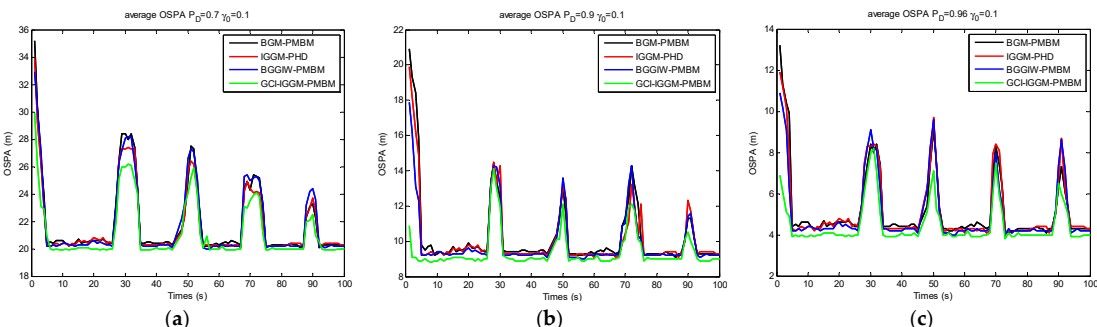

**Figure 4.** OSPA comparisons of GCI–IGGM–PMBM (our proposed scheme) with other multitarget tracking schemes: (**a**) $p_D = 0.7$; (**b**) $p_D = 0.9$; (**c**) $p_D = 0.96$.

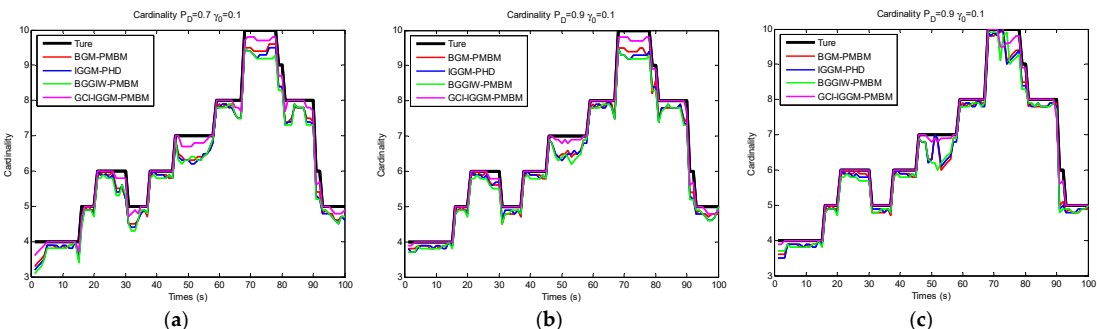

**Figure 5.** Cardinality comparisons of GCI–IGGM–PMBM (our proposed scheme) with other multitarget tracking schemes: (**a**) $p_D = 0.7$; (**b**) $p_D = 0.9$; (**c**) $p_D = 0.96$.

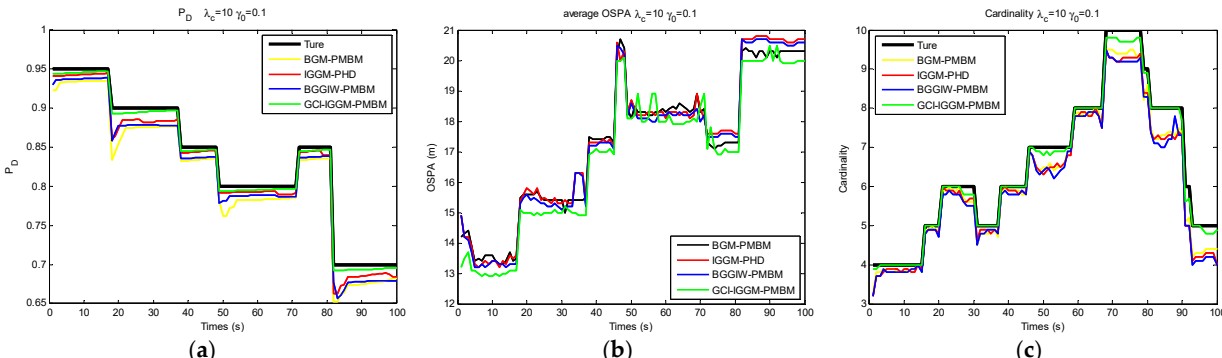

**Figure 6.** Performance comparisons of GCI–IGGM–PMBM with other multitarget tracking schemes with unknown detection probability: (**a**) detection probability estimation; (**b**) OSPA estimation; (**c**) cardinality estimation.

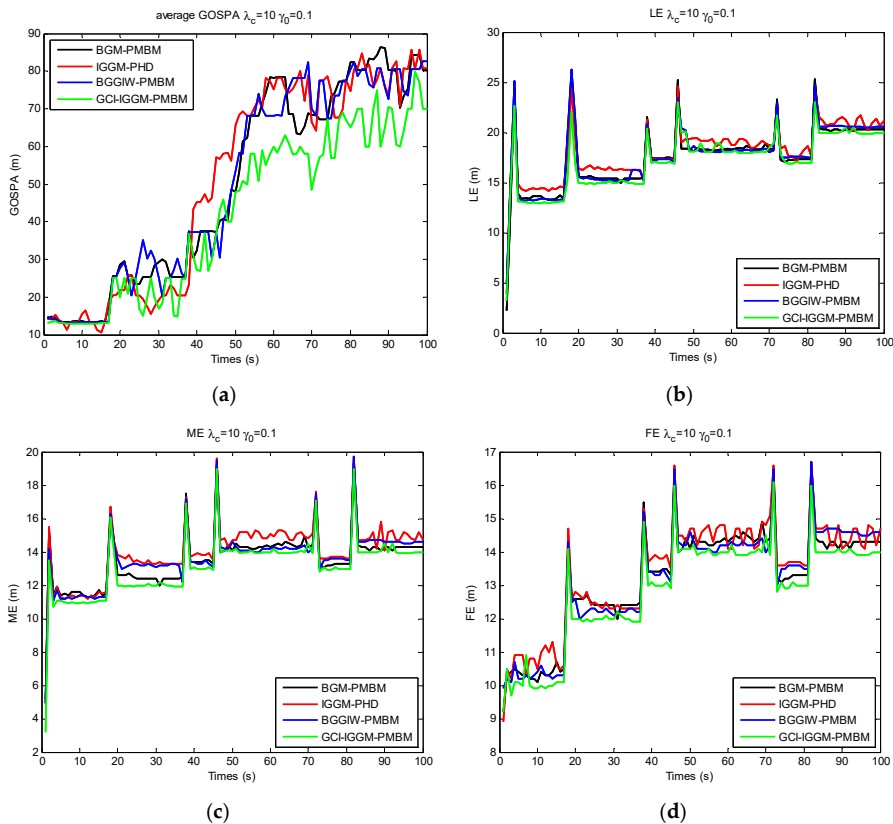

**Figure 7.** Performance comparisons of GCI–IGGM–PMBM with other multitarget tracking schemes with unknown detection probability: (**a**) GOSPA; (**b**) LE; (**c**) ME; (**d**) FE.

The GOSPA distance [36], including the location error (LE) for detected targets, the missed targets error (ME), and false targets error (FE), is shown in Figure 7 for unknown detection probability. It can be noted that the GOSPA, LE, ME, and FE for known detection probabilities ($p_D = 0.7$, $p_D = 0.9$, or $p_D = 0.96$) are not presented in this section. There were abrupt changes for LE, ME, and FE, as shown in Figure 7, at the change junctions of the detection probability, e.g., at 17–18 s, 37–38 s, 48–49 s, 71–72 s, and 82–83 s. The GCI–IGGM–PMBM filter presented relative superiority over other filters except for the BGGIW–PMBM filter at lower detection probability.

Energy consumption is an important metric of multitarget tracking schemes for energy and computation-limited sensor networks, especially for our aquaculture sensor networks. Energy consumption comparisons for these tracking schemes are demonstrated in Figure 8 with unknown detection probability. The energy cost was similar at the stable stage for

each stable detection probability, whereas it changed sharply at the change junctions of detection probability. The anchor rate plays an important role in energy consumption, as more anchors bring about more energy efficiency, because more information is processed more quickly by more anchors, and the initialized positioning stage originating from more anchors is more accurate.

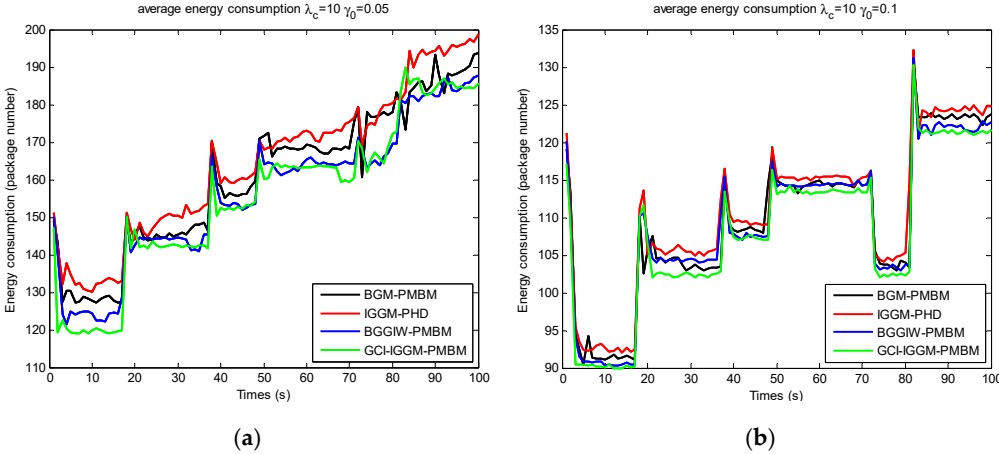

**Figure 8.** Energy consumption comparisons based on anchor rate $\gamma_0$ of GCI–IGGM–PMBM with other multitarget tracking schemes with unknown detection probability: (**a**) $\gamma_0 = 0.05$; (**b**) $\gamma_0 = 0.1$.

### 5.2. Validation of Adaptive Newborn Distribution

Adaptive newborn distribution was proposed in this scheme, as newborn targets can occur at random places in the monitoring area at random periods in actual applications. The OSPA and cardinality comparisons of BGGIW–PMBM and GCI–IGGM–PMBM were derived from the adaptive newborn distribution and known newborn distribution, as shown in Figure 9a. The OSPA tended to relative stable values other than sharp change values after some initialized periods for the two filters with adaptive newborn distribution, whereas it changed sharply with known distribution at the intersections of these trajectories, e.g., at 26–34 s, 46–53 s, 68–74 s, and 88–92 s. The OSPA reached similar values for adaptive and known newborn distributions in other periods. The cardinality presented a similar behavior to that of OSPA, as shown in Figure 9b. The actual number of each scheme slowly reached the true cardinality for the adaptive distribution, whereas the actual number matched the true cardinality after sharp changes at the intersection points.

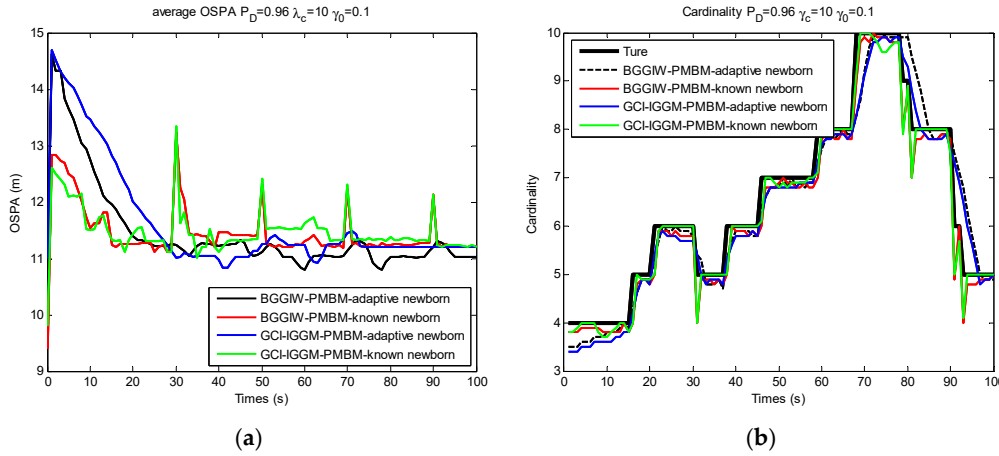

**Figure 9.** OSPA and cardinality comparisons of GCI–IGGM–PMBM filter and BGGIW–PMBM filter based on newborn distribution: (**a**) average OSPA; (**b**) cardinality.

The GOSPA distance, including the LE, ME, and FE, is shown in Figure 10 for unknown newborn and known newborn distributions. There were abrupt changes for LE, ME, and FE for known newborn distributions, as shown in Figure 10, at the trajectory intersection points, e.g., at 26–34 s, 46–53 s, 68–74 s, and 88–92 s, whereas changes were smooth over transition points for adaptive newborn distributions. The GCI–IGGM–PMBM filter presented relative superiority over BGGIW–PMBM at the initialized tracking stage.

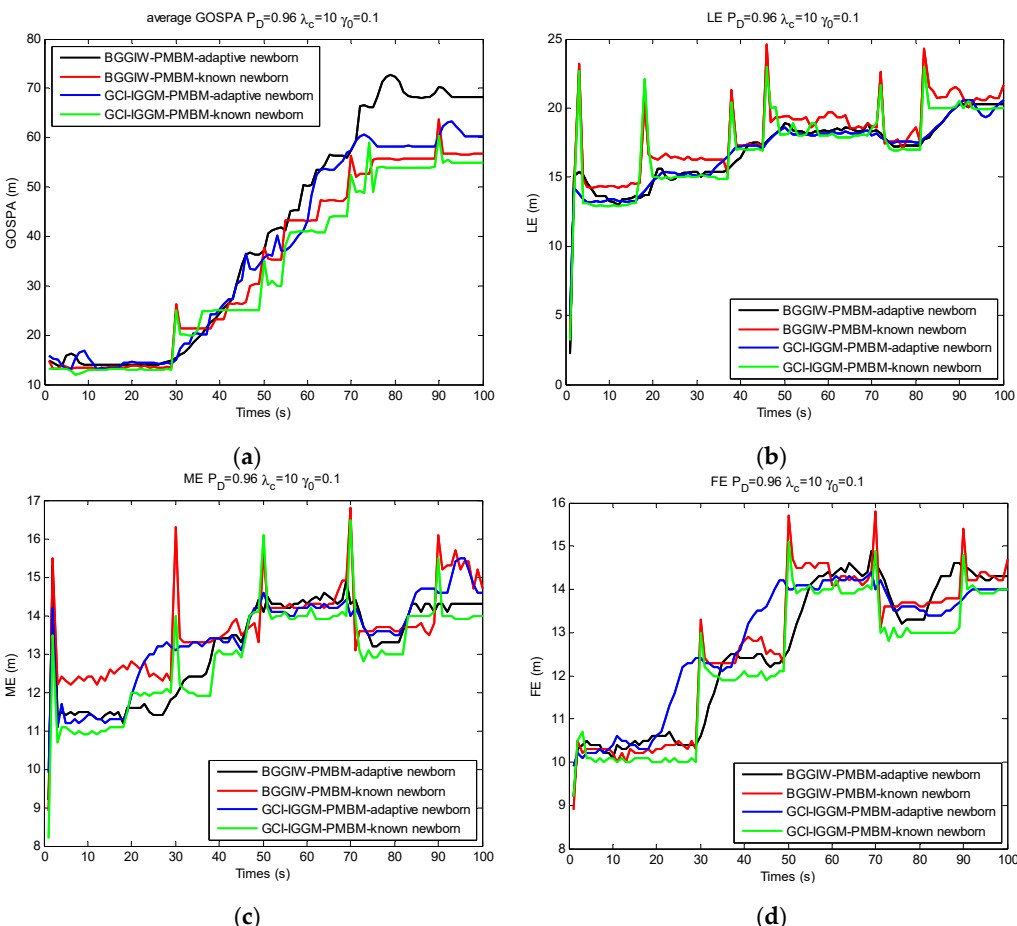

**Figure 10.** Performance comparisons of GCI–IGGM–PMBM and BGGIW–PMBM with newborn distribution: (**a**) GOSPA; (**b**) LE; (**c**) ME; (**d**) FE.

## 6. Conclusions and Future Work

The proposed GCI–IGGM–PMBM filter in this work is aimed at point target tracking, in which multiple targets (e.g., crabs) can be considered as small independent objects or smart points without any spatial extent. The target states and cardinality of multitargets were estimated using the IGGM–PMBM filter. Firstly, the GAM distribution was introduced to present the augmented state of unknown and changing target detection probabilities. Secondly, the intensity of newborn targets was adaptively derived using an inverse gamma (IG) distribution on the basis of this augmented state. On these bases, the detailed recursion and closed-form solutions to the proposed filter were derived by means of approximating the intensity of target birth and potential targets to an IGGM form and the density of existing Bernoulli components to a single IGGM form. Moreover, the associated distributed fusion strategy GCI was applied to a large-scale aquaculture tracking network. Comprehensive experiments were presented to verify the effectiveness of the GCI–IGGM–PMBM tracking method. Comparisons with other multitarget tracking schemes also demonstrated that the tracking behaviors were largely improved; in particular, the tracking energy consumption was reduced sharply, and the tracking accuracy was relatively enhanced.

In our future work, the tracking scheme will take into account not only the positions of objects as in this work, but also the appearance or spatial extent of objects [6]. In addition to crabs, extracting the shape of other farming objects in the farming environment is an important task in tracking multiple targets, which is nearly impossible using point tracking. We will devote ourselves to studying an efficient tracking scheme to resolve this problem, taking various underwater factors into account for multitarget tracking [5]. Moreover, developing smart aquaculture is a general trend for the future, and smart aquaculture can be combined with smart agriculture to form a large agriculture IoT system [37], with many underwater smart devices such as trackers, monitors, or controllers.

**Author Contributions:** Conceptualization, C.L. and J.Z.; methodology, N.X.; software, C.L.; validation, C.L. and J.Z.; formal analysis, C.L. and J.Z.; investigation, Z.T.; resources, Z.T.; data curation, C.L.; writing—original draft preparation, C.L.; writing—review and editing, J.Z.; visualization, C.L.; supervision, N.X.; project administration, Z.T.; funding acquisition, J.Z. All authors read and agreed to the published version of the manuscript.

**Funding:** This research work was supported by the National Natural Science Foundation of China (No. 61362017 and No. 61365007).

**Institutional Review Board Statement:** Not applicable.

**Informed Consent Statement:** Informed consent was obtained from all subjects involved in the study.

**Data Availability Statement:** Not applicable.

**Acknowledgments:** This research work was supported by the Starting Foundation of Shanghai Ocean University (No. A2-0203-00-100344 and No. A2-0203-00-100343). This research work was partly supported by the National Natural Science Foundation of China (No. 61362017 and No. 61365007).

**Conflicts of Interest:** The authors declare no conflict of interest.

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
