# Peer review of "An Improved Multiple-Target Tracking Scheme Based on IGGM–PMBM for Mobile Aquaculture Sensor Networks"

_applsci, doi:10.3390/app13020926_

Round 1

Author Response

     On behalf of my co-authors, we appreciate you very much for the positive and constructive comments and suggestions on our manuscript.   We have studied the comments carefully and have made revision which marked in red in the paper. We have tried our best to revise our manuscript according to the comments. Attached please find the revised version, which we would like to submit for your kind  consideration. We would like to express our great appreciation you for comments on our paper. Looking forward to hearing from you.

Reviewer 2 Report

This paper is interesting and very well written and organized. It has very solid research work like theoretical analysis and simulation etc. The following minor issues should be solved before final publication. 

1. Please carefully proofread this paper, hopefully by native English speakers.

2. Please use same big/small size for all (sub-)section titles, like section 2.1/3.1/6 are different.

3. There are more than 80 equations in this paper which is too much. Please try to reduce to 40 if possible.

4. Please reduce the size of Table 1, Figure 5 etc. as needed.

5. Please give sub-figure title for Fig. 9 a/b/c/d. 

6. Reference part can be improved, and more relevant papers about "MTT, data fusion and eneney efficiency for WSN" is suggested like below.

Jianming Zhang, Juan Sun, Jin Wang, Zongping Li, Xi Chen, An object tracking framework with recapture based on correlation filters and Siamese networks, Computers & Electrical Engineering, 2022. --Zhou, Shu-Ren; Yin, Jian-ping; Zhang, Jian-Ming,Local binary pattern (LBP) and local phase quantization (LBQ) based on Gabor filter for face representation,Neurocomputing, 2013.

Author Response

    On behalf of my co-authors, we appreciate the editor and the reviewers very much for the positive and constructive comments and suggestions on our manuscript. We have studied the comments carefully and have made revision which marked in red in the paper. We have tried our best to revise our manuscript according to the comments. Attached please find the revised version, which we would like to submit for your kind consideration. We would like to express our great appreciation to you for comments on our paper. Looking forward to hearing from you.

Round 2

Author Response

    On behalf of my co-authors, we appreciate the reviewer very much for the positive and constructive comments and suggestions on our  manuscript.

Manuscript ID: applsci-2062551,

Title: An improved multiple target tracking scheme based on IGGM-PMBM for mobile aquaculture sensor networks.

   We have revised the paper carefully according to the comments and adopted the“Marginalia Changes” function in the paper. We would like to express our great appreciation to reviewer for comments on our paper. Looking forward to hearing from you.
